# Joint Distillation for Fast Likelihood Evaluation and Sampling in Flow-based Models

**Xinyue Ai**[2][*][†]**, Yutong He**[1][*]**, Albert Gu**[1]**, Ruslan Salakhutdinov**[1]**, J. Zico Kolter**[1]**,
Nicholas M. Boffi**[1]**, Max Simchowitz**[1]
[1]Carnegie Mellon University, [2]Peking University
yutonghe@cs.cmu.edu, axy25307@stu.pku.edu.cn

## Abstract

Log-likelihood evaluation enables important capabilities in generative models, including model comparison, certain fine-tuning objectives, and many downstream applications. Yet paradoxically, some of today's best generative models – diffusion and flow-based models – still require hundreds to thousands of neural function evaluations (NFEs) to compute a single likelihood. While recent distillation methods have successfully accelerated sampling to just a few steps, they achieve this at the cost of likelihood tractability: existing approaches either abandon likelihood computation entirely or still require expensive integration over full trajectories. We present fast flow joint distillation (F2D2), a framework that simultaneously reduces the number of NFEs required for both sampling and likelihood evaluation by two orders of magnitude. Our key insight is that in continuous normalizing flows, the coupled ODEs for sampling and likelihood are computed from a shared underlying velocity field, allowing us to jointly distill both the sampling trajectory and cumulative divergence using a single flow map. F2D2 is modular, compatible with existing flow-based few-step sampling models, and requires only an additional divergence prediction head. Experiments demonstrate F2D2's capability of achieving accurate log-likelihood with few-step evaluations while maintaining high sample quality, solving a long-standing computational bottleneck in flow-based generative models. As an application of our approach, we propose a lightweight self-guidance method that enables a 2-step MeanFlow to outperform a 1024 step flow matching model with only a single additional backward NFE.

## 1 Introduction

Log-likelihood evaluation and likelihood-based inference have long been fundamental to statistical modeling and machine learning, serving as the backbone for parameter estimation (Fisher, 1922), model selection (Akaike, 1974), and hypothesis testing (Neyman & Pearson, 1933). In the era of generative AI, the ability to efficiently evaluate log-likelihood (log-density) has become even more critical, as it directly enables key post-training techniques including reinforcement learning and preference optimization, where likelihoods are important for methods like PPO, DPO and GRPO (Schulman et al., 2017; Ouyang et al., 2022; Rafailov et al., 2023; Shao et al., 2024). Beyond these applications, optimizing log-likelihood also encourages generative models to capture all modes of the data distribution, avoiding mode collapse that plagues adversarial approaches (Razavi et al., 2019).

While likelihood evaluation is useful for modern generative modeling, the most successful generative models for images and video (Rombach et al., 2022; Black Forest Labs, 2025; OpenAI, 2024; Polyak et al., 2024; Google DeepMind, 2025), namely diffusion and flow matching models, suffer from a critical weakness: computing likelihood requires prohibitively expensive iterative neural function evaluations (NFEs). In particular, discrete-time diffusion models like DDPM (Ho et al., 2020; Nichol & Dhariwal, 2021) require summing up variational bounds across all timesteps, which needs hundreds to thousands of forward passes to compute a single likelihood. Similarly, continuous-time formulations like score SDE (Song et al., 2020) and flow matching (Lipman et al.,

---

[*]Equal contribution.
[†]Work done at CMU.

2022; Albergo & Vanden-Eijnden, 2022; Liu et al., 2022; Albergo et al., 2023) also must integrate the divergence along the learned (probability) flow ODE, which typically requires numerical integration with 100-1000 NFEs for accurate likelihood evaluation. While advanced solvers can significantly reduce NFEs (Karras et al., 2022), they fundamentally cannot escape the integration requirement and produce vastly inaccurate results when restricted to very few steps ($\leq 10$ NFEs). This computational burden makes many likelihood-based finetuning objectives, model comparison, and downstream applications prohibitively expensive for modern diffusion/flow matching models.

Interestingly, diffusion and flow matching models faced the same NFE bottleneck for *sampling* when they were first introduced, where they initially required 1000+ steps to generate a single image. Research addressing this issue has been remarkably successful, with methods that learn to skip multiple steps, either through distillation or self-consistency objectives, emerging as particularly powerful solutions (Salimans & Ho, 2022; Song et al., 2023; Kim et al., 2023; Frans et al., 2024; Geng et al., 2025; Boffi et al., 2025b;a). However, despite achieving few-step sampling, most of these methods completely lose the ability to compute likelihoods, and the few methods that preserve likelihood computation (e.g. Kim et al. (2023)) still require integrating over the entire trajectory with hundreds of NFEs, making fast likelihood evaluation impossible. Thus, while practitioners have solutions to fast sampling for diffusion and flow matching models, fast log-likelihood evaluation remains an unsolved problem. Here, we show that it is possible to achieve *both* at the same time.

**Contributions.** Our key insight is that, in flow matching and continuous normalizing flows (CNFs) in general (Chen et al., 2018), computing exact likelihoods requires solving coupled ODEs: the sampling trajectory $\frac{d}{dt}x_t = v_\theta(x_t, t)$ and the log-density evolution $\frac{d}{dt}\log p_t(x_t) = -\operatorname{div}(v_\theta(x_t, t))$, both depending on the same learned velocity $v_\theta$. Since the divergence term can be viewed as another output derived from the same velocity model, we can learn to distill both the flow trajectory and its corresponding divergence computation simultaneously within a single model. By jointly optimizing for both accurate few-step sampling and log-likelihood evaluation, we can potentially achieve a model that succeeds at both tasks.

Based on these insights, we propose fast flow joint distillation (F2D2), a simple and modular framework for jointly learning fast sampling and fast log-likelihood evaluation in flow-based models. Our key idea is to leverage the flow map framework (Boffi et al., 2025a) to train a single model to predict both the sampling trajectory and cumulative divergence in parallel using a joint self-distillation objective, ensuring both outputs learn to skip the numerous steps in training. This makes F2D2 plug-and-play with any CNF-based few-step sampling method like shortcut models and MeanFlow, and requires only a new divergence prediction head alongside the existing velocity prediction. To our knowledge, F2D2 is the first method to enable accurate few-step log-likelihood evaluation in diffusion/CNF-based generative models, solving a long-standing limitation of these frameworks.

We demonstrate that our method produces both calibrated likelihoods and high quality samples with few-step NFEs on image datasets CIFAR-10 (Krizhevsky et al.) and ImageNet $64 \times 64$ (Deng et al., 2009). We show that our F2D2 are compatible with and can be directly apply to pre-trained shortcut models, MeanFlow and a new distillation method we propose in this paper.

As an application of our method, we introduce maximum likelihood self-guidance, a lightweight test-time intervention that uses rapid likelihood evaluation to optimize over generated samples, requiring only an additional forward and backward pass through the model. Remarkably, we show that F2D2 with maximum likelihood self-guidance instantiated with 2-step MeanFlow achieves lower FID than a 1024-step flow matching model of the same size on CIFAR-10. This proof of concept demonstrates the expanded algorithmic sandbox enabled by rapid likelihood evaluation.

## 2 BACKGROUND

Let $p_{\text{data}}$ denote the data distribution with samples $x \in \mathbb{R}^d$. We consider a time variable $t \in [0, 1]$ where $t = 0$ corresponds to a simple noise distribution $p_0 = \mathcal{N}(0, I)$ and $t = 1$ corresponds to the data distribution $p_1 = p_{\text{data}}$. We denote the marginal distribution at time $t$ as $p_t(x)$.

**Flow Matching.** Flow matching (Lipman et al., 2022; Albergo & Vanden-Eijnden, 2022; Liu et al., 2022; Albergo et al., 2023) is a scalable training method for generative modeling that learns a time-dependent velocity field $v_\theta : \mathbb{R}^d \times [0, 1] \to \mathbb{R}^d$ to transport samples from a simple noise distribution $p_0$ to the data distribution $p_1$. Along a straight-line path $x_t = (1-t)x_0 + tx_1$ that linearly interpolates

between a noise sample $x_0 \sim p_0$ and a data sample $x_1 \sim p_1$, it models the evolution dynamic with the ordinary differential equation ODE:

$$\frac{\mathrm{d}}{\mathrm{d}t}\hat{x}_t = v_\theta(\hat{x}_t, t), \quad x_0 \sim p_0 \tag{2.1}$$

where $\hat{x}_t$ arises from integrating the learned flow model. Since the velocity along this path is simply $x_1 - x_0$, flow matching minimizes the regression objective:

$$\mathcal{L}_{\mathrm{FM}}(\theta) = \mathbb{E}_{t \sim [0,1], x_0 \sim p_0, x_1 \sim p_1} \left[ \|v_\theta(x_t, t) - (x_1 - x_0)\|^2 \right] \tag{2.2}$$

This objective encompasses diffusion models as a special case with different interpolation schemes (Gao et al., 2024). For sampling, it solves the ODE from $t = 0$ to $t = 1$ with numerical solvers like Euler or $\mathrm{dopri5}$, and typically requires 100-1000 NFEs for high quality samples.

**Continuous Normalizing Flows and Likelihood Computation.** A flow-matching model is a special case of a continuous normalizing flow (CNF) (Chen et al., 2018), which transports data from an initial distribution $x_0 \sim p_0$ to an estimated distribution by integrating an ODE. In the case of flow-based models, this is precisely Eq. (2.1). An advantage of the CNF formalism is the ability to explicitly compute likelihoods via the coupled system of ODEs:

$$\frac{\mathrm{d}}{\mathrm{d}t} \begin{bmatrix} \hat{x}_t \\ \log p_{t;\theta}(\hat{x}_t) \end{bmatrix} = \begin{bmatrix} v_\theta(\hat{x}_t, t) \\ -\operatorname{div}(v_\theta(\hat{x}_t, t)). \end{bmatrix} \tag{2.3}$$

Above, $\operatorname{div}(v_\theta(\hat{x}_t, t)) = \operatorname{Tr}(\nabla_{\hat{x}_t} v_\theta(\hat{x}_t, t))$ denotes the divergence of the velocity field $v_\theta$, and $p_{t;\theta}$ represents the likelihood of $\hat{x}_t$ under Eq. (2.1), which we note depends on model parameter via $v_\theta$. Integrating backwards from $t = 1$ (data) to $t = 0$ (noise) with initial conditions $[x_1, 0]^\top$, we obtain:

$$\log p_1(x_1) = \log p_0(\hat{x}_0) + \int_1^0 \operatorname{div}(v_\theta(\hat{x}_t, t))dt = \log p_0(\hat{x}_0) - \int_0^1 \operatorname{div}(v_\theta(\hat{x}_t, t))dt, \tag{2.4}$$

where $\hat{x}_0$ and the intermediate $\hat{x}_t$'s are obtained by integrating the flow backward from $x_1$.

Likelihood evaluation is typically expensive, requiring both careful, finely-discretized integration of an ODE across time steps, and a computation of the divergence term whose exact computation (or the variance of its randomized estimator (Grathwohl et al., 2018)) scales at least linearly in ambient dimension $d$. Thus, likelihood evaluation is far more computationally burdensome than sampling.

**Few-Step Flow-based Models.** To address the computational expense of multiple ODE integrations in *sampling*, recent few-step flow-based models (Kim et al., 2023; Frans et al., 2024; Geng et al., 2025; Boffi et al., 2025a;b) learn to directly predict the outcome of integrating the ODE in Eq. (2.1) using only a small number of function evaluations (NFEs). These methods can be viewed as sharing a common strategy of learning to predict the *flow map* of the underlying ODE.

**Definition 2.1** (Flow Map). Given an ODE $\mathrm{d}x_t = v(x_t, t)\mathrm{d}t$, the flow map $\Phi : \mathbb{R}^d \times [0,1]^2 \to \mathbb{R}^d$ is the solution operator that maps any state at time $t$ to its corresponding state at time $s$:

$$\Phi(x_t, t, s) = x_t + \int_t^s v(x_\tau, \tau)d\tau = x_s \tag{2.5}$$

After learning the flow map with network parameter $\theta$, one can directly perform few-step sampling by discretizing the time interval $[0, 1]$ into $K$ steps with timesteps $0 = t_0 < t_1 < \ldots < t_K = 1$, and iteratively applying the learned flow map: $\hat{x}_{t_{i+1}} = \Phi_\theta(\hat{x}_{t_i}, t_i, t_{i+1})$ for $i = 0, \ldots, K - 1$, starting from $x_0 \sim p_0$. This reduces sampling from hundreds of ODE solver steps to just $K$ NFEs (typically $K < 10$), as each application of $\Phi$ directly predicts the integrated result over the interval $[t_i, t_{i+1}]$ without explicit numerical integration.

## 3 METHOD

We propose to jointly accelerate both sampling and *likelihood evaluation* by learning a flow-map on the joint ODE system described in in Eq. (2.3). Again, $p_0 = \mathcal{N}(0, I)$ is the source distribution, $p_1 = p_{\mathrm{data}}$. $p_t$ represents the marginal distribution of the interpolant $x_t = tx_1 + (1-t)x_0$, $v$ denotes the ground truth velocity and $p_{t;\theta}$ is the distribution of $\hat{x}_t$ under the learned flow model Eq. (2.1). Our aim is to design model which supports two key capabilities:

1. **Fast sampling:** Draw a $\hat{x}_1$ from a trained flow model, using a few number of NFEs, $K_{\text{samp}} < 10$.

2. **Fast likelihood evaluation:** Evaluate the log likelihood of either model samples $\hat{x}_1$ or data samples $x_1$ using a few number of NFEs, $K_{\text{ll}} < 10$.

### 3.1 FAST FLOW JOINT DISTILLATION (F2D2): PARAMETRIZING A JOINT FLOW MAP

Our key insight is that we can apply few-step flow-based models to the ODE in Eq. (2.3) which jointly parametrizes sampling and likelihood evaluation. Following Boffi et al. (2025a); Frans et al. (2024); Geng et al. (2025), we adopt a linear parametrization of the flow map:

$$\Phi_\theta(\hat{x}_t, t, s) = \hat{x}_t + (s - t)u_\theta(\hat{x}_t, t, s) \tag{3.1}$$

where $u_\theta : \mathbb{R}^d \times [0,1]^2 \to \mathbb{R}^d$ predicts the average velocity that directly transports states from time $t$ to time $s$ and ideally $u_\theta(x_t, t, s) \approx \frac{1}{s-t} \int_t^s v(x_\tau, \tau)d\tau$. With this parametrization, we recovers an estimate of the instantaneous velocity as $u_\theta(x_t, t, t)$ in the $s \to t$ limit, and obtain simple conditions for valid flow maps.

**Proposition 3.1** (Flow Map Conditions (Boffi et al., 2025a)). *An operator $\Phi(x, t, s) = x + (s - t)u(x, t, s)$ is a valid flow map if and only if for all $(t, s) \in [0, 1]^2$ and for all $x \in \mathbb{R}^d$, $u(x, t, t) = v(x, t)$ and any of the following conditions holds:*

*(a) $\Phi$ solves the **Lagrangian equation** $\partial_s \Phi(x, t, s) = u(\Phi(x, t, s), s, s)$.*

*(b) $\Phi$ solves the **Eulerian equation** $\partial_t \Phi(x, t, s) + \nabla_x \Phi(x, t, s)u(x, t, t) = 0$.*

*(c) $\Phi$ satisfies the **semigroup property** $\Phi(\Phi(x, t, r), r, s) = \Phi(x, t, s)$ for $t < r < s$.*

Let $z_t = \log p_t(x_t) \in \mathbb{R}$ and $\hat{z}_t = \log p_{t;\theta}(\hat{x}_t) \in \mathbb{R}$ denote the log likelihood, we can then separately parametrize the flow maps for the two subsystems in Eq. (2.3) as

$$\begin{aligned}
\Phi_{X;\theta_X}(\hat{x}_t, t, s) &= \hat{x}_t + (s - t)u_{\theta_X}(\hat{x}_t, t, s), \\
\Phi_{Z;\theta_Z}(\hat{x}_t, \hat{z}_t, t, s) &= \hat{z}_t + (s - t)D_{\theta_Z}(\hat{x}_t, t, s)
\end{aligned} \tag{3.2}$$

Here $u_{\theta_X}(\hat{x}_t, t, s)$ still estimates the average velocity, and $D_{\theta_Z}(x_t, t, s)$ approximates the average divergence $D_{\theta_Z}(x_t, t, s) \approx -\frac{1}{s-t} \int_t^s \text{div}(v(x_\tau, \tau))d\tau$ along the true trajectory between $t$ and $s$.

Notice that average divergence depends *only* on $x_t$, not $z_t$. The fact that $x_t$ is sufficient in our parametrization follows from the joint ODE Eq. (2.3), where the evolution of the likelihood $z_t$ is determined by the divergence of the first flow evaluated at $x_t$.

Therefore, denoting the joint state at time $t$ as $y_t = (x_t, z_t)^\top$, we can then parametrize the joint flow map using shared parameter $\theta$ as

$$\begin{aligned}
\Phi_{Y;\theta}(\hat{y}_t, t, s) &= \begin{bmatrix} \Phi_X(\hat{x}_t, t, s) \\ \Phi_Z(\hat{x}_t, \hat{z}_t, t, s) \end{bmatrix} = \hat{y}_t + (s - t)f_\theta(\hat{x}_t, t, s), \\
f_\theta(\hat{x}_t, t, s) &= \begin{bmatrix} u_\theta(\hat{x}_t, t, s) \\ D_\theta(\hat{x}_t, t, s) \end{bmatrix}
\end{aligned} \tag{3.3}$$

Above, the networks for $u_\theta$ and $D_\theta$ share the same backbone with separate prediction heads for their respective components. The exact architecture is described in Appendix C.

**Theoretical Justification.** To justify the parameterization Eq. (3.3), we recall a property denoted the tangent condition by Boffi et al. (2025a), leveraged by Kim et al. (2023); Geng et al. (2025) to recover the instantaneous velocity and divergence in the $s \to t$ limit:

**Lemma 3.2** (Tangent Condition). *The flow map $\Phi_Y(y, t, s)$ for the joint system Eq. (2.3) satisfies $\lim_{t \to s} \partial_s \Phi_Y(y, t, s) = f(x, s, s) = (v(x, t), -\text{div}(v(x_t, t)))^\top$.*

We can then characterize valid joint flow maps under our parametrization as the following:

**Proposition 3.3** (Characterization of the Joint Flow Map). *Let $\Phi_Y(y, t, s) = y + (s - t)f(x, t, s)$ satisfy $f(x, s, s) = (v(x, t), -\text{div}(v(x_t, t)))^\top$ denotes the dynamics for the joint sampling and likelihood system Eq. (2.3). Then, $\Phi_Y(y, t, s)$ is the flow map for the joint system if and only if $\forall (y, t, s) \in \mathbb{R}^{d+1} \times [0, 1]^2$, any of the following conditions are satisfied:*

---

**Algorithm 1** F2D2 Flow Map Training

---

1: **for** each training step **do**
2:     $x_1 \sim p_{\text{data}}$, $x_0 \sim p_0$, $(t, s) \sim \mathcal{U}([0, 1]^2)$             ▷ forward-only training uses $t \leq s$
3:     $x_t \leftarrow (1 - t)x_0 + tx_1$
4:     Calculate the sampling (velocity) flow map losses $\mathcal{L}_{\text{VM}}(\theta)$ and $\mathcal{L}_{\text{u}}(\theta)$
5:     Calculate the likelihood (divergence) flow map losses $\mathcal{L}_{\text{div}}(\theta)$ and $\mathcal{L}_{\text{D}}(\theta)$
6:     $\mathcal{L}_{\text{F2D2}}(\theta) := \mathcal{L}_{\text{VM}}(\theta) + \mathcal{L}_{\text{u}}(\theta) + \mathcal{L}_{\text{div}}(\theta) + \mathcal{L}_{\text{D}}(\theta)$
7:     Update $\theta$ w.r.t. $\mathcal{L}_{\text{F2D2}}(\theta)$
8: **end for**
9: **return** $\theta$

---

(a) $\Phi_Y$ solves the **Lagrangian equation** $\partial_s \Phi_Y(y, t, s) = f(\Phi_Y(y, t, s), s, s)$.

(b) $\Phi$ solves the **Eulerian equation** $\partial_t \Phi_Y(y, t, s) + \nabla_y \Phi_Y(y, t, s)f(y, t, t) = 0$.

(c) $\Phi$ satisfies the **semigroup property** $\Phi_Y(y, t, s) = \Phi_Y(\Phi_Y(y, t, r), r, s)$ for $t < r < s$.

We provide the full analysis of the characterization in Appendix A. We refer to the family of algorithms which learns a joint map of this characterization as **fast flow joint distillation (F2D2)**. Notably, we can derive four separate training objectives – one pair for the sampling subsystem and the other pair for the likelihood subsystem – and jointly optimizing them yields a valid flow map for Eq. (2.3). The general F2D2 training objective is:

$$\mathcal{L}_{\text{F2D2}}(\theta) := \mathcal{L}_{\text{VM}}(\theta) + \mathcal{L}_{\text{u}}(\theta) + \mathcal{L}_{\text{div}}(\theta) + \mathcal{L}_{\text{D}}(\theta) \tag{3.4}$$

where the first two terms optimize for the sampling flow map $\Phi_X$: $\mathcal{L}_{\text{VM}}$ enforces the instantaneous velocity matching (i.e. the tangent condition, which is often enforced by the flow matching loss in practice), while $\mathcal{L}_{\text{u}}$ enforces one of the flow map conditions from Proposition 3.1 for $\Phi_X$. Similarly, the last two terms optimize for the likelihood flow map $\Phi_Z$: $\mathcal{L}_{\text{div}}$ matches the instantaneous divergence, and $\mathcal{L}_{\text{D}}$ ensures that $\Phi_Z$ satisfies the conditions needed for the joint flow map $\Phi_Y$ to be valid according to the conditions in Proposition 3.3. We provide the pseudocode for the general recipe of F2D2 training in Algorithm 1.

## 3.2 INSTANTIATING F2D2 WITH SHORTCUT AND MEANFLOW

Though our method is, in principle, compatible with any flow map-based method, we instantiate our formulation for Shortcut Models (Frans et al., 2024), based on the semigroup property, and for MeanFlow (Geng et al., 2025), based on the Eulerian equation. We also provide the Lagrangian self-distillation (LSD) (Boffi et al., 2025a) instantiation based on the Lagrangian equation in Appendix A. Pseudocode for each instantiation is provided in Appendix C.

### 3.2.1 JOINT SHORTCUT: SHORTCUT-F2D2

Shortcut models (Frans et al., 2024) enforce the semigroup property (Proposition 3.1 (c)) by applying it to the midpoint between timesteps $t$ and $s$. This amounts to the shortcut self-consistency loss:

$$\mathcal{L}_{\text{u-SC}}(\theta) = \mathbb{E}_{t,s,x_t} \left[ \left\| u_\theta(x_t, t, s) - \frac{1}{2}\text{sg}\left(u_\theta(x_t, t, r) + u_\theta(\Phi_{X;\theta}(x_t, t, r), r, s)\right) \right\|^2 \right] \tag{3.5}$$

where $\text{sg}(\cdot)$ denotes stop-gradient. Combined with the tangent condition (at $t = s$),

$$v_\theta(x, t) = u_\theta(x, t, t), \tag{3.6}$$

and training with the flow matching loss as $\mathcal{L}_{\text{VM}}$,

$$\mathcal{L}_{\text{VM-SC}}(\theta) := \mathbb{E}_{t \sim [0,1], x_0 \sim p_0, x_1 \sim p_1} \left[ \left\| u_\theta(x_t, t, t) - (x_1 - x_0) \right\|^2 \right] \tag{3.7}$$

this self-consistency loss $\mathcal{L}_{\text{u-SC}}$ serves as $\mathcal{L}_{\text{u}}$ and yields a valid flow map by enforcing both the tangent condition and the semigroup property (see Corollary B.1 in Appendix B for details).

We convert this method to a joint self-distillation method by introducing the two additional losses:

$$\mathcal{L}_{\text{div-SC}}(\theta) := \mathbb{E}\left[ \left\| D_\theta(x_t, t, t) + \text{div}(u_\theta^-(x_t, t, t)) \right\|^2 \right] \tag{3.8}$$

$$\mathcal{L}_{\text{D-SC}}(\theta) := \mathbb{E}_{t < s} \left[ \left\| D_\theta(x_t, t, s) - \frac{1}{2}\text{sg}\left(D_\theta(x_t, t, r) + D_\theta(\Phi_{X;\theta}(x_t, t, r), r, s)\right) \right\|^2 \right] \tag{3.9}$$

The first loss proceeds by analogy to Eq. (3.6), where the correct $D_\theta$ is precisely the instantaneous velocity field in the $Z$-component, and this is precisely $\text{div}(u_\theta)$ by Eq. (2.3). The second loss simply the enforces the semigroup property on the $Z$-component. Take then together, we arrive at the shortcut variant of F2D2, Shortcut-F2D2, obtained by minimizing the loss

$$\mathcal{L}_{\text{SC-F2D2}}(\theta) := \mathcal{L}_{\text{VM-SC}}(\theta) + \mathcal{L}_{\text{u-SC}}(\theta) + \mathcal{L}_{\text{div-SC}}(\theta) + \mathcal{L}_{\text{D-SC}}(\theta). \tag{3.10}$$

We demonstrate the derivation of Shortcut-F2D2 in Appendix A and provide the pseudocode for Shortcut-F2D2 in Algorithm 3 and 4.

### 3.2.2 JOINT MEANFLOW: MEANFLOW-F2D2

Alternatively, MeanFlow (Geng et al., 2025) learns the time-averaged velocity $u_\theta(x, t, s)$ by enforcing the so-called MeanFlow identity $u_\theta(x_t, s, t) = v(x_t, t) + (s-t)\frac{d}{dt}u_\theta(x_t, s, t)$, which we show in Corollary B.2 in Appendix B solves the Eulerian equation (Proposition 3.1 (b)). To our knowledge, this is the first proof of this fact.

For efficiency, $\frac{d}{dt}u(x, s, t)$ can be computed as $\partial_t u(x, t, s) + \nabla_x u(x, t, s)v(x, t)$ obtained via a Jacobian–vector product (JVP). The training objective is then to enforce this identity by regressing the model's prediction to the target implied thereby:

$$\mathcal{L}_{\text{MF}}(\theta) = \mathbb{E}_{t,s,x_0,x_1}\left[\left\|u_\theta(x_t, t, s) - \text{sg}\left((s-t)(\partial_t u(x_t, t, s) + \nabla_x u(x_t, t, s)v(x_t, t)) + v(x_t, t)\right)\right\|^2\right], \tag{3.11}$$

where $v(x_t, t) = x_1 - x_0$ is the ground truth instantaneous velocity of the interpolant. Importantly, Eq. (3.11) encapsulates both $\mathcal{L}_{\text{VM}}$ and $\mathcal{L}_{\text{u}}$ since $\mathcal{L}_{\text{MF}}$ recovers the flow matching loss when $s = t$.

To extend this this method to the joint system, we introduce the additional loss

$$\mathcal{L}_{\text{div-MF}}(\theta) = \mathbb{E}\left[\left\|D_\theta(x_t, t, s) - \text{sg}((s-t)(\partial_t D_\theta(x_t, t, s)\right.\right. \tag{3.12}$$
$$\left.\left. + \nabla_x D_\theta(x_t, t, s)v(x_t, t)) - \text{div}(u_\theta(x_t, t, t)))\right\|^2\right],$$

By analogy to $\mathcal{L}_{\text{MF}}$, $\mathcal{L}_{\text{div-MF}}$ obviates the need for explicit divergence matching term $\mathcal{L}_{\text{div}}$. MeanFlow-F2D2 then amounts to training the objective

$$\mathcal{L}_{\text{MF-F2D2}}(\theta) = \mathcal{L}_{\text{MF}}(\theta) + \mathcal{L}_{\text{div-MF}}(\theta). \tag{3.13}$$

We demonstrate the derivation of MeanFlow-F2D2 in Appendix A and provide the pseudocode for Shortcut-F2D2 in Algorithm 5.

### 3.3 PRACTICAL DESIGN CHOICES

While instantaneous velocity supervision for sampling is straightforward to obtain from data, obtaining reliable and tractable supervision for the instantaneous divergence presents significant challenges. We address these through several key practical considerations.

**Parameter Sharing.** Since both $X$ and $Z$ components of our joint flow map derive from the same underlying velocity field, learning to predict both components simultaneously is fundamentally learning two transformations of the same underlying dynamics. As a result, we efficiently parametrize the joint flow map using a shared backbone network with two separate prediction heads. This parameter sharing architecture ensures both outputs are derived from a consistent representation of the flow dynamics and reduces the number of parameters compared to training separate models.

**Hutchinson Trace Estimator.** Computing divergence terms $\text{div}(v(x_t, t))$ requires $O(d)$ backward passes for exact computation, making training prohibitively expensive for high-dimensional data. Following standard practice (Grathwohl et al., 2018; Lipman et al., 2022; Song et al., 2020), we employ the Hutchinson trace estimator $\text{div}(v) \approx \mathbb{E}_{\epsilon \sim \mathcal{N}(0,I)}[\epsilon^\top \nabla_x v \cdot \epsilon]$ which provides unbiased estimates with only $O(1)$ computational cost per training step.

**Staged Training with Warm Start.** Since divergence supervision depends on having accurate velocity predictions, we adopt a staged training approach. In practice, we pre-train the sampling velocity component $u_\theta$ alone using existing flow map distillation techniques, which provides a good initialization for joint training later. Optionally, we can also pre-train a teacher flow matching model $v_\phi$ that serves as a reliable source of divergence supervision, replacing the potentially noisy

---

**Algorithm 2** Maximum Likelihood Self-Guidance Sampling with F2D2

1: $x_0 \sim p_0$
2: $D \leftarrow D_\theta(x_0, 0, 1)$
3: $\mathcal{L}_{\mathrm{NLL}} \leftarrow -\log p_0(x_0) - D$
4: $x_0 \leftarrow \mathsf{Adam}(x_0, \mathcal{L}_{\mathrm{NLL}})$          ▷ One step Adam update w.r.t. $x_0$ optimizing $\mathcal{L}_{\mathrm{NLL}}$
5: $t_0, \ldots, t_{K_{\mathrm{samp}}} \leftarrow \mathsf{linspace}(0, 1, K_{\mathrm{samp}} + 1)$
6: **for** $i = 0, \ldots, K_{\mathrm{samp}} - 1$ **do**
7:      $u \leftarrow u_\theta(x_i, t_i, t_{i+1})$
8:      $x_{i+1} \leftarrow x_i + (t_{i+1} - t_i)u$
9: **end for**
10: **return** $x_1$

---

$\mathrm{div}(u_\theta(x_t, t, t))$ with the more accurate $\mathrm{div}(v_\phi(x_t, t))$ during joint distillation. When using a pre-trained teacher model or warm-starting, we replace the flow matching loss with a teacher matching loss, which will be described next.

**Shortcut-Distill-F2D2.** To further improve training stability and performance, we propose Shortcut-Distill, a shortcut model variant that combines the semigroup flow map with a learned teacher instantaneous velocity. Our three-stage pipeline consists of: (1) **Teacher pre-training:** Train $v_\phi$ using standard flow matching; (2) **Shortcut-Distill:** Warm start $\theta$ with the teacher parameters and replace $\mathcal{L}_{\mathrm{div\text{-}sc}}$ with teacher supervision: $\mathbb{E}_t \left[ \|u_\theta(x_t, t, t) - v_\phi(x_t, t)\|^2 \right]$; (3) **Joint distillation:** Warm start $\theta$ from sampling distillation, add divergence head and train both components jointly. This approach maintains the semigroup condition while leveraging a pre-trained velocity field to ensure the joint flow map are well-aligned.

**Adaptations to forward-only training.** While shortcut models and MeanFlow are both consistent with the full flow-map formulation, their practical use (i.e. sampling) and the available pretrained checkpoints are forward-only: training and tuning assume $t \leq s$. Several implementation choices are exclusively designed for this regime (e.g., the specific parameters of MeanFlow's logit-normal time sampling, shortcut models' discretized time sampling schedule, and common logarithmical time-parameterizations used in EDM/CTM-style models). Extending training to explicitly cover both $t \leq s$ and $t \geq s$ would require re-deriving or re-tuning these design choices and can break the plug-and-play compatibility with existing pretrained models and algorithms.

In practice, we use a simple first-order approximation to reuse a forward-only model for backward likelihood integration. Let $\Delta x(x_t, t, t + \Delta t) = \Delta t\, u_\theta(x_t, t, t + \Delta t)$ denote the forward predicted displacement. We approximate the backward displacement by $\Delta x(x_t, t, t - \Delta t) \approx -\Delta x(x_t, t, t + \Delta t)$ and apply the same approximation to the log-density increment. This approximation is first-order accurate in $\Delta t$ and works well empirically at small-to-moderate step sizes. While there exist out-of-distribution inputs at $t = 1$, in practice we find that the networks can generalize to these scenarios with forward-only training.

### 3.4 APPLICATION: MAXIMUM LIKELIHOOD SELF-GUIDANCE WITH F2D2

Now that we have access to log-likelihood computation with few NFEs, we can explore various applications. One particularly interesting one is using the one-step divergence prediction (combined with the source distribution's log-likelihood) as a pseudo-likelihood objective for inference-time optimization. Specifically, we can optimize the initial noise $x_0$ to improve sample quality before running the sampling procedure. This approach resembles reward-based initial noise optimization for one-step generation models (Eyring et al., 2024), except we do not require external reward models. Instead, we obtain the guidance signal from the model's own likelihood prediction head – effectively performing self-guidance at inference time to improve sample quality. This maximum likelihood self-guidance sampling algorithm is described in Algorithm 2.

## 4 RELATED WORKS

**Likelihood computation in diffusion and flow models.** While diffusion and flow-based models excel at sample generation, their likelihood evaluation remains computationally expensive.

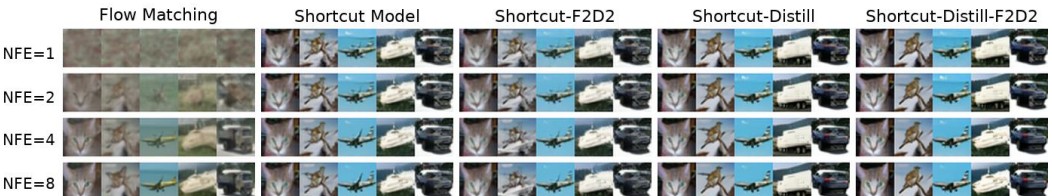

Figure 1: CIFAR-10 generated samples from different models with different numbers of steps.

Discrete-time diffusion models compute likelihoods through variational bounds requiring hundreds of NFEs (Ho et al., 2020; Nichol & Dhariwal, 2021). Continuous formulations enable exact likelihood via the probability flow ODE (Song et al., 2020) but require numerical integration with 100-1000 NFEs. Prior research has explored various techniques to improve likelihood estimation (Grathwohl et al., 2018; Song et al., 2021), but they still require many NFEs for accurate evaluation. In parallel, normalizing flows (Rezende & Mohamed, 2015) offer exact and tractable log-likelihoods by using specialized network architecture designs and change-of-variables formula. Many recent efforts (Zhai et al., 2024; Ho et al., 2019; Chen et al., 2019) aim at scaling up the normalizing flow principles through better dequantization and and advance architectures.

**Accelerating sampling in flow-based models.** Reducing sampling costs has been a major focus in diffusion and flow matching research. Advanced ODE solvers (Karras et al., 2022; Lu et al., 2022) leverage the semi-linear structure of the probability flow ODE to reduce discretization error. Distillation methods (Salimans & Ho, 2022; Sauer et al., 2024) and self-distillation models (Song & Dhariwal, 2023; Zhou et al., 2025) provide alternative solutions by training student models to match teacher trajectories or training to perform self-bootstrapping with fewer steps. In particular, consistency models (Song et al., 2023) and consistency trajectory models (Kim et al., 2023) learn direct mappings from any point along the trajectory to data.

**Flow map-based methods.** Flow maps provide a general framework to model the solution operator of ODEs, enabling direct prediction of integrated trajectories (Kim et al., 2023; Boffi et al., 2025a;b). Recent works exploit this structure for few-step sampling. For example, as we have shown above, shortcut models (Frans et al., 2024) imposes semigroup property to learn the flow maps while MeanFlow (Geng et al., 2025) and Align Your Flow (Sabour et al., 2025) enforce Eulerian conditions. While these methods successfully reduce sampling to less than 10 NFEs, they either abandon likelihood computation entirely or still require full trajectory integration for likelihood evaluation.

## 5 EXPERIMENTS

### 5.1 SETUPS

We empirically verify the effectiveness of our method on image datasets CIFAR-10 (Krizhevsky et al.), ImageNet $64 \times 64$ (Deng et al., 2009) and CelebA-64 (Liu et al., 2015). We evaluate the sample quality using Fréchet Inception Distance (FID) (Heusel et al., 2017) on 50K generated images. The negative log-likelihood (NLL) is measured in bits per dimension (BPD) on the entire test set of CIFAR-10 and CelebA-64 and a randomly sampled 10K subset of the ImageNet test set. We compare our method against flow matching (Lipman et al., 2022), shortcut models (Frans et al., 2024), Lagrangian self-distillation (LSD) (Boffi et al., 2025a) and MeanFlow (Geng et al., 2025) as baselines, and augment the later three for joint distillation. All models are unconditionally trained. We use $1, 2, 4, 8$ Euler steps for sampling and likelihood evaluation. In order to facilitate fair comparison to the baselines, all shortcut and MeanFlow models are trained with $t \leq s$ while LSD models are trained with the full $(t, s) \sim \mathcal{U}([0, 1]^2)$. Implementation details can be found in Appendix C.

### 5.2 RESULTS

**CIFAR-10** Table 1 and Figure 1 show the quantitative and qualitative comparison on CIFAR-10 respectively. As we can observe, flow matching yields poor FID and invalid NLL estimates in few-step setting. Shortcut model and MeanFlow achieve significantly better FID and are able to compute NLL

Table 1: NLL and FID results on CIFAR-10 dataset with different numbers of Euler steps. The flow matching model here, which achieves BPD 3.12 as the NLL with 1024 steps and FID 2.60 with 200 steps, is also the teacher model we use in our Shortcut-Distill. For NLL, the closer to the teacher result (3.12 BPD) the better, and for FID, the lower the better. We denote the best results in **bold**, the second best with underlines, the overall best results in ⎡boxes⎤ and invalid predictions in gray color.

| Method | 8 Steps | | 4 Steps | | 2 Steps | | 1 Step | |
|---|---|---|---|---|---|---|---|---|
| | NLL | FID | NLL | FID | NLL | FID | NLL | FID |
| Flow Matching | -9.93 | 20.63 | -24.01 | 64.27 | -52.85 | 146.24 | -111.19 | 313.54 |
| Shortcut Model | -12.07 | 7.10 | -28.03 | 9.63 | -60.01 | 16.04 | -124.15 | 27.28 |
| Shortcut-Distill (Ours) | -11.42 | 5.01 | -26.82 | 5.41 | -57.72 | 7.13 | -119.42 | 12.75 |
| MeanFlow | -9.00 | 4.34 | -21.26 | 5.14 | -46.63 | 2.84 | -97.59 | **2.80** |
| Shortcut-F2D2 (Ours) | 3.07 | 8.78 | **3.26** | 10.21 | **2.73** | 15.58 | 0.20 | 27.35 |
| Shortcut-Distill-F2D2 (Ours) | ⎡3.12⎤ | 5.68 | 2.87 | 5.96 | 2.38 | 7.35 | 1.62 | 13.76 |
| MeanFlow-F2D2 (Ours) | 2.38 | **3.78** | 1.34 | **4.37** | 1.63 | ⎡2.59⎤ | **3.51** | 3.02 |

Table 2: Negative log-likelihood (NLL) measured in BPD and FID results on ImageNet 64×64 dataset with different numbers of Euler steps. The flow matching model here, which achieves BPD 3.34 as the NLL with 1024 steps and FID 13.09 with 200 steps, is also the teacher model we use in our Shortcut-Distill. For NLL, the closer to the teacher result (3.34 BPD) the better, and for FID, the lower the better. We denote the best results in **bold** and invalid predictions in gray color.

| Method | 8 Steps | | 4 Steps | | 2 Steps | | 1 Step | |
|---|---|---|---|---|---|---|---|---|
| | NLL | FID | NLL | FID | NLL | FID | NLL | FID |
| Flow Matching | -6.41 | 31.60 | -15.87 | 68.55 | -35.23 | 170.00 | -74.54 | 363.39 |
| Shortcut-Distill (Ours) | -9.03 | **19.47** | -22.30 | **21.73** | -49.01 | **28.12** | -102.07 | **42.72** |
| Shortcut-Distill-F2D2 (Ours) | **3.51** | 21.91 | **3.94** | 24.05 | **3.97** | 29.83 | **1.54** | 44.02 |

for their ability to recover instantaneous velocity, but their NLL values remain invalid. Incorporating our proposed F2D2 brings the NLL estimations to a calibrated range close to the teacher's BPD across different settings. In particular, both Shortcut-F2D2 and Shortcut-Distill-F2D2 substantially improves NLL compared to plain their orignal counterparts while maintaining competitive FID, indicating that F2D2 can provide reasonable likelihood estimates without sacrificing much sample quality. Finally, MeanFlow-F2D2 shows FID improvements relative to the original MeanFlow while simultaneously producing calibrated NLL, demonstrating that F2D2's potential in providing complementary training signals that are beneficial to both components.

**ImageNet** 64 × 64 Shown in Table 2, flow matching quickly degenerates under few-step sampling, with invalid NLL and extremely poor FID. Shortcut-Distill improves the few-step FID but still produces invalid NLL. By contrast, Shortcut-Distill-F2D2 achieves both competitive FID and meaningful likelihoods close to the teacher's BPD of 3.34 across all step counts. These results further confirm F2D2's ability for simultaneous fast sampling and fast likelihood evaluation.

**CelebA-64** As shown in Table 3, the flow matching model degenerates under few-step Euler sampling, yielding invalid likelihood estimates and worsening sample quality. LSD substantially improves the few-step FID, but still produce invalid NLL across all step counts. In contrast, LSD-F2D2 achieves both superior image quality and well-calibrated likelihoods in a few steps.

## 5.3 Maximum Likelihood Self-guidance with MeanFlow-F2D2

Figure 2 shows the FID and qualitative comparison among different methods built upon MeanFlow using the number of forward NFE on CIFAR-10. As we can observe, our F2D2 improves the model's inference time scaling ability. With additional self-guidance, the model not only surpasses the baseline MeanFlow performance but also outperforms a 1024-step flow matching model of the same size, demonstrating the effectiveness its own likelihood predictions as valid signals to guide the sampling process toward higher-quality generations.

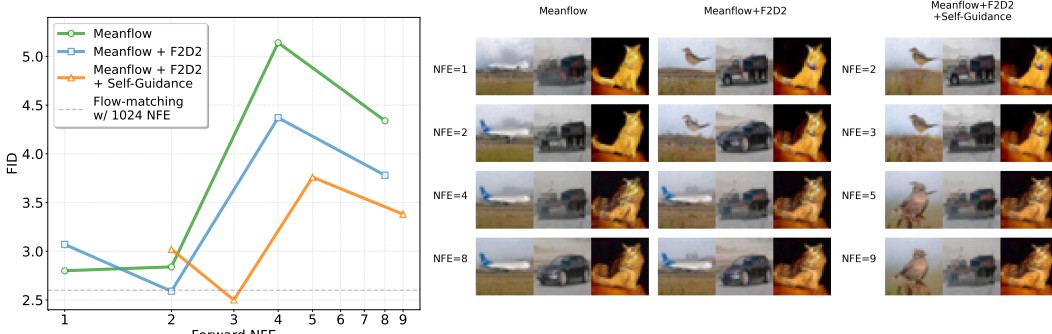

(a) FID comparison across different numbers of forward NFEs.

(b) Example samples from various MeanFlow-based models using different numbers of forward NFEs.

Figure 2: Results of MeanFlow-based methods on CIFAR-10.

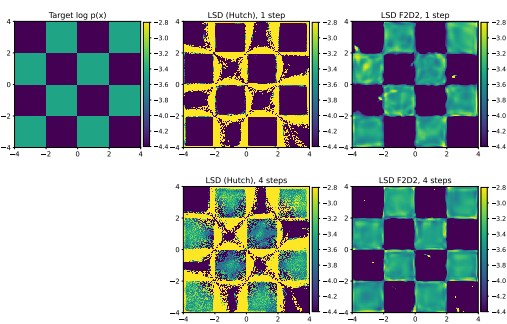

Figure 3: LSD-based log-likelihood evaluation comparison on 2D checkerboard dataset.

Table 3: Negative log-likelihood (NLL) measured in BPD and FID results on CelebA-64 dataset with different numbers of Euler steps. The flow matching model here achieves BPD 1.75 in 1024 steps and FID 2.48 in 200 steps. For NLL, closer to the flow matching estimate is better; for FID, lower is better. Best in **bold**; invalid in gray.

| Method | 8 Steps | | 4 Steps | | 2 Steps | | 1 Step | |
|---|---|---|---|---|---|---|---|---|
| | NLL | FID | NLL | FID | NLL | FID | NLL | FID |
| Flow Matching | -6.88 | 30.60 | -16.39 | 58.14 | -36.46 | 120.65 | -77.51 | 181.23 |
| LSD | -6.78 | 3.33 | -14.89 | 4.04 | -32.72 | 6.32 | -69.83 | 12.96 |
| LSD-F2D2 (Ours) | **1.64** | **2.41** | **1.75** | **2.75** | **1.73** | **3.86** | **1.64** | **6.94** |

## 5.4 2D CHECKERBOARD

In this section, we present a set of comparison of log-likelihood evaluation results on 2D checkerboard, a synthetic dataset with analytically tractable ground truth log-likelihood. As we can observe in Figure 3, without F2D2, vanilla LSD catastrophically fails at few-step log-likelihood estimation. On the other hand, our LSD-F2D2 is able to accurately recover the target density distribution even with only 1 NFE, preserving both spatial structure and density values. This directly validates that, while traditional methods fail at few-step likelihood evaluation, F2D2 enables accurate likelihood via joint flow map distillation.

## 6 CONCLUSION

We present fast flow joint distillation (F2D2), a simple and modular framework that enables both fast sampling and fast likelihood evaluation in flow-based generative models. By jointly distilling the sampling trajectory and divergence computation into a unified flow map, our method simultaneously achieves accurate likelihood evaluation and high sample quality with just a few NFEs. Our experiments on CIFAR-10 and ImageNet $64 \times 64$ demonstrate that F2D2 maintains accurate likelihood estimates while preserving sample quality when applied to existing few-step methods including shortcut models and MeanFlow. The efficiency gains from F2D2 enable new algorithmic possibilities, as illustrated by our maximum likelihood self-guidance method, which enables a 2-step MeanFlow model to outperform a 1024-step flow matching model of the same size on CIFAR-10. As flow-based models continue to scale, we believe that efficient likelihood evaluation alongside fast sampling will become increasingly important for enabling new training objectives, model analysis techniques, and downstream applications that require both capabilities.

## REPRODUCIBILITY STATEMENT

We provide proofs to our theoretical results in Appendix A and B. We also provide the implementation details to reproduce our algorithm and experimental results in Section 5 and Appendi C. The links to the PyTorch and JAX implementations of our algorithm can be found on our website: https://kellyyutonghe.github.io/f2d2/.

## ACKNOWLEDGMENT

MS would like to recognize a TRI University 2.0 Research Partnership and a Google Robotics Research Award. We also thank Michael S. Albergo and Eric Vanden-Eijnden for the helpful discussions.

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

# A  CHARACTERIZATION OF THE JOINT FLOW MAP

In this section, we provide an analytical characterization of the joint flow map for the combined sampling and likelihood dynamics Eq. (2.3). For the simplicity of notations, we define the joint system

$$\frac{\mathrm{d}}{\mathrm{d}t}x_t = v(x_t, t), \qquad\qquad x(0) = x_0 \sim \mathcal{N}(0, I)$$
$$\frac{\mathrm{d}}{\mathrm{d}t}z_t = -\mathrm{div}(v(x_t, t)), \qquad z_0 = \log p_0(x_0) \tag{A.1}$$

where we will use the shorthand $z_t$ for $\log p_t(x_t)$. Moreover, we define the right-hand side of Eq. (A.1) as $g(y, t)^\top = (v(x, t), -\mathrm{div}(v(x_t, t)))^\top$ where $y = (x, z)^\top$.

We then define the flow maps for the two subsystems in Eq. (A.1) as

$$\Phi_X(x, t, s) = x + (s - t)u(x, t, s),$$
$$\Phi_Z(x, z, t, s) = z + (s - t)D(x, t, s) \tag{A.2}$$

In Eq. (A.2), we note that the hierarchical structure in Eq. (A.1) is explicit, and that the function $D$ only depends on $x$ and not on $z$. We then define the joint flow map as

$$\Phi_Y(y, t, s) = \begin{bmatrix} \Phi_X(x, t, s) \\ \Phi_Z(x, z, t, s) \end{bmatrix} = y + (s - t)f(x, t, s),$$
$$f(x, t, s) = \begin{bmatrix} u(x, t, s) \\ D(x, t, s) \end{bmatrix} \tag{A.3}$$

We first recall the simple tangent identity denoted the tangent condition by Boffi et al. (2025a), also leveraged by (Kim et al., 2023; Frans et al., 2024; Geng et al., 2025), which allows us to recover the instantaneous velocity and divergence in the $s \to t$ limit:

**Lemma A.1** (Tangent Condition). *The flow map $\Phi_Y(y, t, s)$ for the joint system Eq. (A.1) satisfies* $\lim_{t \to s} \partial_s \Phi_Y(y, t, s) = g(y, s) = f(x, s, s)$. *In particular, $u(x, s, s) = v(x, t)$ and $D(x, s, s) = -\mathrm{div}(v(x, s))$.*

*Proof.* The proof follows by application of Lemma 2.1 from Boffi et al. (2025a). □

Now, given Eq. (A.3), we may now state the following proposition, which is based on an identity similar to Lemma A.1 in reverse.

**Proposition A.2** (Characterization of the Joint Flow Map). *Let $\Phi_Y(y, t, s) = y + (s - t)f(x, t, s)$ satisfy $f(x, s, s) = g(y, s)$ where $g(y, t)^\top = (v(x, t), -\mathrm{div}(v(x_t, t)))^\top$ denotes the dynamics for the joint sampling and likelihood system Eq. (A.1). Then, $\Phi_Y(y, t, s)$ is the flow map for the joint system if and only if any of the following conditions are satisfied:*

1. *(Lagrangian condition) $Z_{s,t}$ satisfies the Lagrangian equation*

$$\partial_s \Phi_Y(y, t, s) = f(\Phi_Y(y, t, s), s, s) \qquad \forall \ (y, t, s) \in \mathbb{R}^{d+1} \times [0, 1]^2. \tag{A.4}$$

2. *(Eulerian condition) $\Phi_Y(y, t, s)$ satisfies the Eulerian equation*

$$\partial_t \Phi_Y(y, t, s) + \nabla_y \Phi_Y(y, t, s)f(y, t, t) = 0, \qquad \forall \ (y, t, s) \in \mathbb{R}^{d+1} \times [0, 1]^2. \tag{A.5}$$

3. *(Semigroup condition) $Z_{s,t}$ satisfies the semigroup property*

$$\Phi_Y(y, t, s) = \Phi_Y(\Phi_Y(y, t, r), r, s), \qquad \forall \ (y, t, r, s) \in \mathbb{R}^{d+1} \times [0, 1]^3, t < r < s. \tag{A.6}$$

*Proof.* The proof follows by application of Proposition 2.2 from Boffi et al. (2025a) applied to the joint system. □

Proposition A.2 gives three characterizations of the joint flow map that may be used to devise few-step flow-based training algorithms. In each case, by definition of Eq. (A.3), the $X$ block reduces to the flow map characterizations introduced in Boffi et al. (2025b;a) for the sampling system. The second block for the likelihood dynamics is new, which we focus on now to instantiate the resulting equations.

**Lagrangian likelihood equation.** By inspection, Eq. (A.4) leads to the equation

$$D(x, t, s) = -\text{div}(v(\Phi_X(x, t, s), s)) - (s - t)\partial_s D(\Phi_X(x, t, s), t, s). \tag{A.7}$$

Squaring the residual leads to the objective function

$$\mathcal{L}(\hat{D}) = \mathbb{E}\left[\left\|\hat{D}(x_t, t, s) - (-\text{div}(v(\Phi_X(x_t, t, s), s)) - (s - t)\partial_s D(\Phi_X(x_t, t, s), t, s))\right\|^2\right] \tag{A.8}$$

At training time, because we don't have access to the true $-\text{div}(v)$ or the true sampling flow map $\Phi_X(x, t, s)$, we may replace them by their self-consistent estimates,

$$\mathcal{L}(\theta) = \mathbb{E}\left[\left\|D_\theta(x_t, t, s) - \text{sg}\left(-\text{div}(u_\theta(\Phi_{X;\theta}(x_t, t, s), s, s)) - (s - t)\partial_s D_\theta(\Phi_{X;\theta}(x_t, t, s), t, s)\right)\right\|^2\right] \tag{A.9}$$

where we have also placed a stopgrad operator to avoid backpropagation through Jacobian-vector products and to control the flow of information from the teacher to the student. This gives the Lagrangian likelihood self-distillation algorithm.

Notice that in the limit as $t \to s$, the Lagrangian condition can cover the tangent condition.

**Eulerian likelihood equation.** To derive our Eulerian schemes, we first note that

$$\nabla_y \Phi_Y(y, t, s) = \begin{bmatrix} \nabla_x \Phi_X(x, s, t) & 0 \\ \nabla_x \Phi_Z(x, z, t, s) & \nabla_z \Phi_Z(x, z, t, s) \end{bmatrix}. \tag{A.10}$$

Hence to compute the second component of the Eulerian equation, we must collect some simple algebraic identities,

$$\begin{aligned} \partial_t \Phi_Z(x, z, t, s) &= -D(x, t, s) + (s - t)\partial_t D(x, t, s), \\ \nabla_x \Phi_Z(x, z, t, s) &= (s - t)\nabla_x D(x, t, s), \\ \nabla_z \Phi_Z(x, z, t, s) &= I. \end{aligned} \tag{A.11}$$

Using the above, we find that the Eulerian relation for $Y$ becomes

$$-D(x, t, s) + (s - t)\partial_t D(x, t, s) + (s - t)\nabla_x D(x, t, s)v(x, t) - \text{div}(v(x, t)) = 0. \tag{A.12}$$

We may enforce this equation by minimizing the square residual,

$$\mathcal{L}(\hat{D}) = \mathbb{E}\left[\left\|D(x_t, t, s) - ((s - t)\partial_t D(x_t, t, s) + (s - t)\nabla_x D(x_t, t, s)v(x, t) - \text{div}(v(x_t, t)))\right\|^2\right]. \tag{A.13}$$

At training time, we again place a $\text{sg}(\cdot)$ operator to avoid backpropagating through the derivatives,

$$\mathcal{L}(\theta) = \mathbb{E}\left[\left\|D_\theta(x_t, t, s) - \text{sg}\left((s - t)\partial_t D_\theta(x_t, t, s) + (s - t)\nabla_x D_\theta(x_t, t, s)v(x, t) - \text{div}(v(x_t, t))\right)\right\|^2\right]. \tag{A.14}$$

In the above, we do not have access to the ideal $v(x_t, t)$ nor $-\text{div}(v(x_t, t))$. However, we observe that because of the placement of $\text{sg}(\cdot)$, resulting gradient will be linear in $v(x_t, t)$, so that we may replace it by its Monte Carlo estimate. In practice, this reduces to conditional-OT flow matching. Second, we replace $-\text{div}(v(x_t, t))$ by the self-consistent estimate $-\text{div}(u_\theta(x_t, t, t))$, leading to

$$\mathcal{L}(\theta) = \mathbb{E}\left[\left\|D_\theta(x_t, t, s) - \text{sg}\left((s - t)\partial_t D_\theta(x_t, t, s) + (s - t)\nabla_x D_\theta(x_t, t, s)(x_1 - x_0) - \text{div}(u_\theta(x_t, t, t))\right)\right\|^2\right]. \tag{A.15}$$

This gives the MeanFlow-F2D2 algorithm.

**Semigroup property.** Last, we consider the semigroup approach. The second block is given by,

$$Y_{s,t}(z) = Y_{u,t}(Y_{s,u}(z)). \tag{A.16}$$

$$\Phi_Y(y, t, s) = \Phi_Y(\Phi_Y(y, t, r), r, s) \tag{A.17}$$

Writing this out using Eq. (A.3), we find that

$$\begin{aligned} z + (s - t)D(t, s) &= \Phi_Z(x, z, t, r) + (s - r)D(\Phi_Z(x, z, t, r), r, s), \\ \iff z + (s - t)D(x, t, s) &= z + (r - t)D(x, t, r) + (s - r)D(\Phi_X(x, t, r), r, s), \end{aligned} \tag{A.18}$$

Setting $r = \frac{1}{2}(t+s)$ recovers a continuous limit of shortcut models, as shown in Boffi et al. (2025a). In this case, Eq. (A.18) becomes

$$D(x,t,s) = \frac{1}{2}\left(D(x,t,r) + D(\Phi_X(x,t,r),r,s)\right), \tag{A.19}$$

Squaring the residual gives

$$\mathcal{L}(\hat{D}) = \mathbb{E}\left[\left\|\hat{D}(x_t,t,s) - \frac{1}{2}\left(D(x_t,t,r) + D(\Phi_X(x_t,t,r),r,s)\right)\right\|^2\right], \tag{A.20}$$

where as in the above Eulerian and Lagrangian approaches, we have replaced the ideal flow map $X$ by the self-consistent estimate. Again, to control the flow of information we may place a stopgrad,

$$\mathcal{L}(\theta) = \mathbb{E}\left[\left\|D_\theta(x_t,t,s) - \frac{1}{2}\mathsf{sg}\left(D_\theta(x_t,t,r) + D_\theta(\Phi_{X;\theta}(x_t,t,r),r,s)\right)\right\|^2\right], \tag{A.21}$$

which gives our Shortcut-F2D2 algorithm.

## B  ADDITIONAL THEORETICAL ANALYSIS

**Corollary B.1**. *Shortcut models enforce semigroup property with their self-consistency loss.*

*Proof.* The same proof as the semigroup property for $\Phi_Y$ holds for $\Phi_X$. $\qquad\square$

**Corollary B.2**. *MeanFlow (Geng et al., 2025) directly enforces the Eulerian condition with Mean-FLow identity.*

*Proof.* Since $u(x_t,t,t) = v(x_t,t)$ is implied by the linear parametrization in Equation 3.1, we can have

$$\partial_t \Phi(x_t,t,s) + \nabla_{x_t}\Phi(x_t,t,s)u(x_t,t,t) = 0$$
$$\partial_t(x_t + (s-t)u(x_t,t,s)) + \nabla_{x_t}(x_t + (s-t)u(x_t,t,s))v(x_t,t) = 0$$
$$-u(x_t,t,s) + (s-t)\partial_t u(x_t,t,s) + (I + (s-t)\nabla_{x_t}u(x_t,t,s))v(x_t,t) = 0$$
$$(s-t)\left(\partial_t u(x_t,t,s) + \nabla_{x_t}u(x_t,t,s)v(x_t,t)\right) + v(x_t,t) = u(x_t,t,s)$$
$$(s-t)\frac{d}{dt}u(x_t,t,s) + v(x_t,t) = u(x_t,t,s)$$

which is exactly MeanFlow identity. $\qquad\square$

|  | **CIFAR-10** | **ImageNet-**$64 \times 64$ | **CelebA-64** |
|---|---|---|---|
| Noise embedding | Positional | Positional | Positional |
| Channels | 128 | 192 | 128 |
| Channels multiple | 1,2,2,2 | 1,2,3,4 | 1,2,3,4 |
| Attention resolution | 16 | 32,16,8 | 16,8 |
| Residual blocks | 4 | 3 | 3 |
| Dropout | 0.13 | 0.1 | 0.0 |
| Batch size | 512 | 1024 | 256 |
| GPUs | 8 L40S | 8 L40S | 4 L40S |
| Iterations | 150k | 125k | 350k |
| Learning Rate | 1e-4 (constant) | 1e-4 (constant) | 1e-2 (Sqrt decay at 35k) |
| Warmup Steps | 30k (linear warmup) | 60k (linear warmup) | - |
| Precision | float32 | float32 | bfloat16 |
| Optimizer | RAdam | RAdam | RAdam |
| EMA rate | 0.9999 | 0.9999 | 0.9999 |

Table 4: Training hyperparameters for flow-matching.

## C    IMPLEMENTATION DETAILS

In this section, we provide the training details for each model experimented in this paper, and the pseudocode for each F2D2 instantiation.

**Flow-matching.** We use linear interpolation to generate training targets. Flow-matching models are trained on both unconditional CIFAR-10 and ImageNet-$64 \times 64$ using the configurations in Table 4, with 200 standard Euler steps for sampling. This model serves as the teacher model for the second distillation stage.

**Shortcut Model.** We reimplement the shortcut model ( Frans et al. (2024)) for CIFAR-10. Following their method, we strictly sample $s \geq t$ (i.e. forward-only training) and use discrete timesteps and set $1/128$ as the smallest unit of time for approximating the ODE. We consider 8 possible shortcut lengths ranging from $(1, 1/2, \dots, 1/128)$. Similarly, we divide the batch into $3/4$ for flow-matching training and $1/4$ for self-consistency training. However, instead of DiT, we use a U-Net backbone with configurations in Table 4. We train for 100k iterations on CIFAR-10 and sample with 1, 2, 4, and 8 standard Euler steps. Additionally, we follow the original paper and parametrize the model to take $x_t$, $t$ and $s - t$ as inputs.

**Shortcut-Distill.** We use the velocity predicted by the flow-matching model as the target for the flow-matching loss, instead of $x_{\text{data}} - x_{\text{noise}}$. All other configurations remain the same as in the Shortcut Model.

**MeanFlow.** We directly use the pre-trained model weights and parametrization provided by the official PyTorch repository to conduct the CIFAR-10 MeanFlow experiments. The provided pre-trained MeanFlow model has the same size as all other models we implement for CIFAR-10. Because the pre-trained MeanFlow models is forward only (i.e. $t \leq s$), we follow the same convention and adapt our algorithm based on the considerations that we have elaborated in Section 3.

**LSD.** We follow the experimental setup of Boffi et al. (2025a). We allocate $75\%$ of each batch to the flow matching loss and $25\%$ to the self-distillation loss for the first 200k iterations. And then we allocate $50\%$ of each batch to the flow matching loss and $50\%$ to the self-distillation loss for another 150k iterations. To enable both forward and backward estimation, we sample two time steps uniformly at random for the self-distillation loss, without enforcing any ordering constraint between them (i.e., we do not require one to be larger than the other). All LSD models are trained with the full $(t, s) \sim \mathcal{U}([0, 1]^2)$.

**Log-likelihood estimation with vanilla flow map models.** Since Flow maps can recover instantaneous velocity through their tangent condition $u_\theta(x_t, t, t) \approx v(x_t, t)$, flow map models trained only

---

**Algorithm 3** Shortcut-F2D2 Training

---

1: **for** each training step **do**
2:     $x_1 \sim p_{\text{data}}, x_0 \sim p_0, (t, s) \sim \mathcal{U}([0,1]^2)$                    ▷ forward-only training uses $t \leq s$
3:     $r \leftarrow (t + s)/2$
4:     $x_t \leftarrow (1 - t)x_0 + tx_1$
5:     $x_r \leftarrow x_t + (r - t)u_\theta(x_t, t, r)$
6:     $\mathcal{L}_{\text{VM-SC}}(\theta) \leftarrow \|u_\theta(x_t, t, t) - (x_1 - x_0)\|^2$
7:     $\mathcal{L}_{\text{u-SC}}(\theta) \leftarrow \|u_\theta(x_t, t, s) - \frac{1}{2}\text{sg}\left(u_\theta(x_t, t, r) + u_\theta(x_r, r, s)\right)\|^2$
8:     $D_t \leftarrow \text{sg}\left(\text{div}(u_\theta(x_t, t, t))\right)$
9:     $\mathcal{L}_{\text{div-SC}}(\theta) \leftarrow \|D_\theta(x_t, t, t) + D_t\|^2$
10:     $\mathcal{L}_{\text{D-SC}}(\theta) \leftarrow \|D_\theta(x_t, t, s) - \frac{1}{2}\text{sg}\left(D_\theta(x_t, t, r) + D_\theta(x_r, r, s)\right)\|^2$
11:     $\mathcal{L}_{\text{SC-F2D2}}(\theta) \leftarrow \mathcal{L}_{\text{VM-SC}}(\theta) + \tilde{\mathcal{L}}_{\text{u-SC}}(\theta) + \mathcal{L}_{\text{div-SC}}(\theta) + \mathcal{L}_{\text{D-SC}}(\theta)$
12:     Update $\theta$ w.r.t. $\mathcal{L}_{\text{SC-F2D2}}(\theta)$
13: **end for**
14: **return** $\theta$

---

**Algorithm 4** Shortcut-Distill-F2D2 Training

---

1: **for** each training step **do**
2:     $x_1 \sim p_{\text{data}}, x_0 \sim p_0, (t, s) \sim \mathcal{U}([0,1]^2)$                    ▷ forward-only training uses $t \leq s$
3:     $r \leftarrow (t + s)/2$
4:     $x_t \leftarrow (1 - t)x_0 + tx_1$
5:     $x_r \leftarrow x_t + (r - t)u_\theta(x_t, t, r)$
6:     $\mathcal{L}_{\text{VM-SC}}(\theta) \leftarrow \|u_\theta(x_t, t, t) - v_\phi(x_t, t)\|^2$                    ▷ match with flow matching teacher $v_\phi$
7:     $\mathcal{L}_{\text{u-SC}}(\theta) \leftarrow \|u_\theta(x_t, t, s) - \frac{1}{2}\text{sg}\left(u_\theta(x_t, t, r) + u_\theta(x_r, r, s)\right)\|^2$
8:     $D_t \leftarrow \text{sg}\left(\text{div}(v_\phi(x_t, t))\right)$                    ▷ matching teacher divergence
9:     $\mathcal{L}_{\text{div-SC}}(\theta) \leftarrow \|D_\theta(x_t, t, t) + D_t\|^2$
10:     $\mathcal{L}_{\text{D-SC}}(\theta) \leftarrow \|D_\theta(x_t, t, s) - \frac{1}{2}\text{sg}\left(D_\theta(x_t, t, r) + D_\theta(x_r, r, s)\right)\|^2$
11:     $\mathcal{L}_{\text{SC-F2D2}}(\theta) \leftarrow \mathcal{L}_{\text{VM-SC}}(\theta) + \tilde{\mathcal{L}}_{\text{u-SC}}(\theta) + \mathcal{L}_{\text{div-SC}}(\theta) + \mathcal{L}_{\text{D-SC}}(\theta)$
12:     Update $\theta$ w.r.t. $\mathcal{L}_{\text{SC-F2D2}}(\theta)$
13: **end for**
14: **return** $\theta$

---

for sampling can, in principle, evaluate likelihood by computing $div(u_\theta(x_t, t, t))$ and solving for Eq 2.3 and 2.4. As a result, we use Euler solver with this formuation to produce the NLL estimation of the baseline Shorcut, Shortcut-Distill and MeanFlow models.

**F2D2 for Shortcut Model and Shortcut-Distill.** For F2D2 training, we add a scalar head after the UNet decoder. The scalar head is implemented as an MLP: the decoder's final feature map is first flattened, then passed through two fully connected layers with SiLU activations, and finally projected to a single scalar. This head maps the spatial feature representation into a scalar value, which we use to predict divergence.

For CIFAR-10, the hidden sizes are 128 and 64, while for ImageNet-$64 \times 64$ they are 64 and 16. We linearly warm-start the model with teacher weights and train it using the flow-matching loss, with the teacher model taken from either the Shortcut Model or the Shortcut Model-Distill. We also adopt the same discrete timesteps as in the Shortcut Model. To balance the four losses, we scale down the divergence distillation targets by a factor of 20,000 for CIFAR-10 and 300,000 for ImageNet-$64 \times 64$. We train Shortcut-Distill-F2D2 for 10k iterations on CIFAR-10, 10k iterations on ImageNet-$64 \times 64$. We also train Shortcut-F2D2 for 10k iterations on CIFAR-10. We stop the training when the calibrated value of bpd is reached. All other configurations remain the same. Although our derivation matches $-\text{div}(v)$, our implementation parametrizes to predict $\text{div}(v)$, which yields an equivalent formulation as long as the training loss and the sampling process as the appropriate signs.

The pseudocode for Shortcut-F2D2 is provided in Algorithm 3 and the one for Shortcut-Distill-F2D2 is provided in Algorithm 4.

**F2D2 for MeanFlow.** We use the same scalar head for CIFAR-10 with Shortcut-Distill-F2D2 and train for an additional 50 epochs using the same configurations as in the original PyTorch imple-

---

**Algorithm 5** MeanFlow-F2D2 Training

---

1: **for** each training step **do**
2:     $x_1 \sim p_{\text{data}}, x_0 \sim p_0, (t, s) \sim \mathcal{U}([0, 1]^2)$                  ▷ forward-only training uses $t \leq s$
3:     $x_t \leftarrow (1 - t)x_0 + tx_1$
4:     $\mathcal{L}_{\text{MF}}(\theta) = \|u_\theta(x_t, t, s) - \mathsf{sg}\left((s - t)(\partial_t u(x_t, t, s) + \nabla_x u(x_t, t, s)v(x_t, t)) + v(x_t, t)\right)\|^2$
5:     $D_{t,s} \leftarrow \mathsf{sg}((s - t)(\partial_t D_\theta(x_t, t, s) + \nabla_x D_\theta(x_t, t, s)v(x_t, t)) - \text{div}(u_\theta(x_t, t, t)))$
6:     $\mathcal{L}_{\text{div-MF}}(\theta) \leftarrow \|D_\theta(x_t, t, s) - D_{t,s}\|^2$
7:     $\mathcal{L}_{\text{MF-F2D2}}(\theta) \leftarrow \mathcal{L}_{\text{MF}}(\theta) + \mathcal{L}_{\text{div-MF}}(\theta)$
8:     Update $\theta$ w.r.t. $\mathcal{L}_{\text{MF-F2D2}}(\theta)$
9: **end for**
10: **return** $\theta$

---

**Algorithm 6** LSD-F2D2 Training

---

1: **for** each training step **do**
2:     $x_1 \sim p_{\text{data}}, x_0 \sim p_0, (t, s) \sim \mathcal{U}([0, 1]^2)$
3:     $x_t \leftarrow (1 - t)x_0 + tx_1$
4:     $x_s \leftarrow x_t + (s - t)u_\theta(x_t, t, s)$
5:     $\mathcal{L}_{\text{LSD}}(\theta) = \|\partial_s x_s - \mathsf{sg}\left(u_\theta(x_s, s, s)\right)\|^2$
6:     $\mathcal{L}_{\text{div-LSD}}(\theta) \leftarrow \|D_\theta(x_t, t, s) - \mathsf{sg}\left(-\text{div}(u_\theta(x_s, s, s)) - (s - t)\partial_s D_\theta(x_s, t, s)\right)\|^2$
7:     $\mathcal{L}_{\text{LSD-F2D2}}(\theta) \leftarrow \mathcal{L}_{\text{LSD}}(\theta) + \mathcal{L}_{\text{div-LSD}}(\theta)$
8:     Update $\theta$ w.r.t. $\mathcal{L}_{\text{LSD-F2D2}}(\theta)$
9: **end for**
10: **return** $\theta$

---

mentation. We use the pre-trained flow matching model to provide the instantaneous divergence supervision. Algorithm 5 shows the pseudo-code for MeanFlow-F2D2 training.

**F2D2 for LSD.** For F2D2 training, we also add an MLP scalar head for divergence prediction and the hidden sizes are 32 and 8. We change the learning rate to 1e-4 and use $50\%$ of each batch to the flow matching loss and $50\%$ to the self-distillation loss. To balance the velocity and divergence losses, we scale down the divergence distillation targets by a factor of 10,000. We train LSD-F2D2 for 60k iterations. All other configurations remain the same as LSD. Algorithm 6 shows the pseudo-code for LSD-F2D2 training.

**Maximum Likelihood Self-Guidance with F2D2.** We first predict the negative likelihood using randomly sampled noise with 1-step divergence prediction and take this negative likelihood as the loss. We then update the noise with one step of Adam, using a learning rate of $1 \times 10^{-3}$ for 1-step sampling and $5 \times 10^{-3}$ for 2-, 4-, and 8-step sampling. We then evaluate FID using 1, 2, 4, and 8 standard Euler steps for sampling.

**BPD.** We use the same method as Lipman et al. (2022) to compute the BPD. To compute BPD with F2D2 models, we discretize the interval from 1 to 0 into equal segments, while the uniform step size serves as the second time input.

**2D Checkerboard.** We follow the experimental setup of Boffi et al. (2025a). The model is a 4-layer MLP with 512 hidden units per layer and GELU activations. Both the flow-matching baseline and the Shortcut model are trained for 150k iterations with a batch size of 100k and an initial learning rate of $10^{-3}$, using a square-root decay schedule after 35k steps. For the Shortcut model, each batch is divided in a $3 : 1$ ratio between the flow-matching loss and the self-distillation loss. For Shortcut-F2D2, we extend the output layer with an additional dimension and initialize training from the pretrained Shortcut model, running an extra 27k iterations. As for the LSD model, we train it for 200k iterations with a batch size of 10k while keeping all other configurations unchanged. For LSD-F2D2, we add an additional MLP head (hidden size 512) to predict divergence and scale the divergence targets by a factor of $0.1$. In each batch, we use a $1 : 1$ split between the flow-matching loss and self-distillation loss. We train LSD-F2D2 for 200k iterations using a learning rate of $1 \times 10^{-4}$ with a square-root decay schedule starting after 50k steps. All other configurations are kept the same.

Table 5: Sampling speed (wall-clock time and per-image latency) for different NFEs.

| Dataset | Hardware | 8 NFE | 4 NFE | 2 NFE | 1 NFE |
|---|---|---|---|---|---|
| CIFAR-10 | 4×L40S | 146 s (2.92 ms/img) | 83 s (1.66 ms/img) | 51 s (1.02 ms/img) | 40 s (0.80 ms/img) |
| ImageNet-64×64 | 8×L40S | 289 s (5.78 ms/img) | 160 s (3.20 ms/img) | 95 s (1.90 ms/img) | 63 s (1.26 ms/img) |

**Computation Cost and Runtime.** We conduct all our training on an 8-GPU L40S node. To train a typical F2D2 CIFAR10 model, it takes around 1 day for the Flow-matching teacher, 1 day for the vanilla Shortcut or Shortcut-Distill baselines, and 5 hours for the F2D2 finetuning on top of it. For ImageNet64x64 models, it takes around 8 days for the flow matching teacher, 6 days for the vanilla shortcut or shortcut-distill model, and 5 hours for the F2D2 finetuning. For inference run time, we provide the wall-clock time to generate 50k samples for each dataset using different numbers of NFEs in Table 5.

# D  ADDITIONAL ABLATIONS

In this section, we present additional ablation study to further investigate the bevavior of our F2D2 models.

With variance around $1.8 \times 10^6$ each time step, the Hutchinson estimator does produce values that are significantly larger in magnitude than the sampling range. Therefore, as we have mentioned in the Appendix C, we follow common deep learning practices and apply a $5 \times 10^{-5}$ scaling factor to the divergence target in order for the neural network to better learn the prediction. Here in Table 6, we present an ablation study on the divergence scaling factor to validate this choice of hyperparameter. As we can observe, appropriate scaling enables simultaneous calibrated likelihood prediction and high sample quality, while scaling too strong can produce invalid NLL and scaling too weak can degrade overall performance.

Figure 4 shows the self-guidance sampling algorithm with multi-step Adam optimization. As we can observe, while adding optimization steps does not further improve the FID, the performance also does not collapse. We are interested in other variants of our self-guidance algorithm to better improve the performance as exciting future work directions.

Finally, we conduct additional study on the effect that different loss components have on the sample quality as well as likelihood estimation accuracy. In particular, we slightly modify Equation 3.10 to add an additional weighting scalar $\lambda$ to $\mathcal{L}_{\text{div-sc}} + \mathcal{L}_{\text{D-sc}}$ and compare the performance under various weighting.

$$\mathcal{L}_{\text{sc-F2D2}}(\theta) := \mathcal{L}_{\text{VM-sc}}(\theta) + \mathcal{L}_{\text{u-sc}}(\theta) + \lambda(\mathcal{L}_{\text{div-sc}}(\theta) + \mathcal{L}_{\text{D-sc}}(\theta)) \qquad (\text{D.1})$$

Table 7 shows the Shortcut-Distill-F2D2 results with $\lambda = 0.01, 0.1, 1, 10$, with $\lambda = 1$ equivalent to our original setting and the performance of Shortcut-Distill without F2D2 as reference. As we can observe, adding an additional loss weighting can potentially help F2D2 achieve a better sweet spot in balancing sample quality and likelihood estimation: While our original equal weighting loss ($\lambda = 1$) obtains the most calibrated likelihood, it slightly penalizes the FID scores. Similarly, significantly down-weighting ($\lambda = 0.01$) further improves FID but loses the ability to produce accurate likelihood. However, when choosing an appropriate $\lambda$ (in particular when $\lambda = 0.1$), this additional weighting can enable the model to produce both FID scores that match or even slightly improve the original Shortcut-Distill performance, and relatively calibrated log-likelihood estimation.

The experiment in Table 7 opens the door to many intriguing future research directions that can potentially further improve the performance. These include more fine-grained loss weighting scheme and developing training schedules similar to the ones proposed in Frans et al. (2024) and Geng et al. (2025), where the flow matching objective and the self-consistency objective are optimized in separate portions of the training.

Table 6: Ablation study on the divergence scaling factor.

| Scaling | 8 Steps | | 4 Steps | | 2 Steps | | 1 Step | |
|---|---|---|---|---|---|---|---|---|
| | NLL | FID | NLL | FID | NLL | FID | NLL | FID |
| $5 \times 10^{-4}$ | 3.27 | **5.16** | 2.43 | 6.31 | 1.26 | **7.06** | -2.92 | **13.16** |
| $5 \times 10^{-5}$ | **3.12** | 5.68 | **2.87** | **5.96** | **2.38** | 7.35 | **1.62** | 13.76 |
| $5 \times 10^{-6}$ | 5.38 | 15.57 | 5.44 | 25.20 | 5.50 | 10.96 | 5.60 | 23.18 |

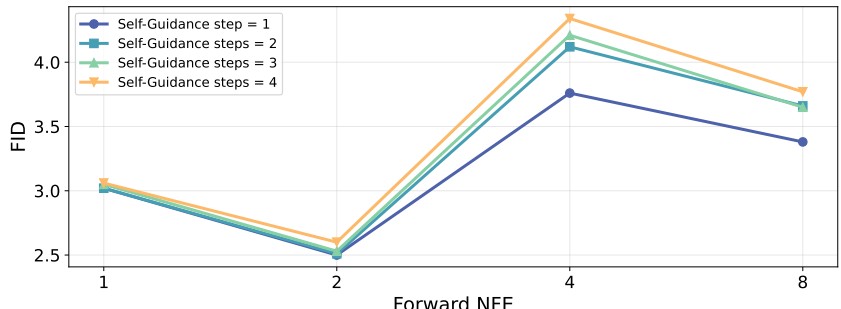

Figure 4: FID results of multi-step self-guidance sampling on CIFAR10.

Table 7: Ablation study on different loss weighting on $\mathcal{L}_{\text{div-sc}} + \mathcal{L}_{\text{D-sc}}$ in Shortcut-Distill-F2D2.

| Scaling | 8 Steps | | 4 Steps | | 2 Steps | | 1 Step | |
|---|---|---|---|---|---|---|---|---|
| | NLL | FID | NLL | FID | NLL | FID | NLL | FID |
| Shortcut-Distill | -11.42 | 5.01 | -26.82 | 5.41 | -57.72 | 7.13 | -119.42 | 12.75 |
| 10 | 1.47 | 11.66 | 0.82 | 11.23 | -0.07 | 10.11 | -2.19 | 18.27 |
| 1 | **3.12** | 5.68 | **2.87** | 5.96 | **2.38** | 7.35 | **1.92** | 13.76 |
| 0.1 | 2.59 | 5.01 | 2.00 | 5.33 | 1.78 | 7.01 | 0.68 | 13.04 |
| 0.01 | 2.04 | **4.87** | 1.51 | **5.28** | 0.59 | **6.76** | -1.68 | **12.70** |

# E    ADDITIONAL RESULTS

In this section, we present additional experimental results. Specifically, in Table 8 we showcase the NLL per sample error w.r.t. the teacher prediction on CIFAR10. As we can observe, the F2D2 variants lower the error by 7-171× in comparison to the baselines. This evaluation directly verifies that our F2D2 can produce sample-level calibrated likelihood estimations that are suitable for practical applications, not merely matching the summary statistics.

Figure 7 demonstrate the training dynamics of our Shortcut-Distill-F2D2 model with the training loss curve, where we can observe stable training with expected small fluctuations native to uniformly random timestep selection at each iteration.

We also provide additional 2D checkerboard results in Figure 5 and 6 and qualitative image generation results in Figure 8,9,10,11,12,13,14,15,16,17,18,19,20,21,22. In particular, Figure 5 shows the generated sample density via forward integration and Figure 6 shows the data sample density via backward integration using different models. All images shown in this section are non-cherry picked results.

Table 8: NLL Mean absolute per sample error w.r.t the teacher's prediction in BPD on CIFAR10. It shows that the F2D2 variants lower the error by 7-171× in comparison to the baselines.

| Method | 8 steps | 4 steps | 2 steps | 1 step |
|---|---|---|---|---|
| Shortcut | 15.11 | 30.97 | 62.85 | 126.58 |
| Shortcut-Distill | 14.47 | 29.79 | 60.61 | 121.99 |
| Meanflow | 12.02 | 24.24 | 49.33 | 100.03 |
| Shortcut-F2D2 | 0.52 | 0.55 | 1.01 | 3.08 |
| Shortcut-Distill-F2D2 | **0.41** | **0.50** | **0.90** | 1.69 |
| MeanFlow-F2D2 | 0.80 | 1.78 | 1.51 | **0.60** |

Table 9: NLL Mean absolute per sample error w.r.t the teacher's prediction in BPD on CelebA-64.

| Method | 8 steps | 4 steps | 2 steps | 1 step |
|---|---|---|---|---|
| LSD | 8.45 | 16.59 | 34.46 | 71.64 |
| LSD-F2D2 | **0.18** | **0.24** | **0.33** | **0.53** |

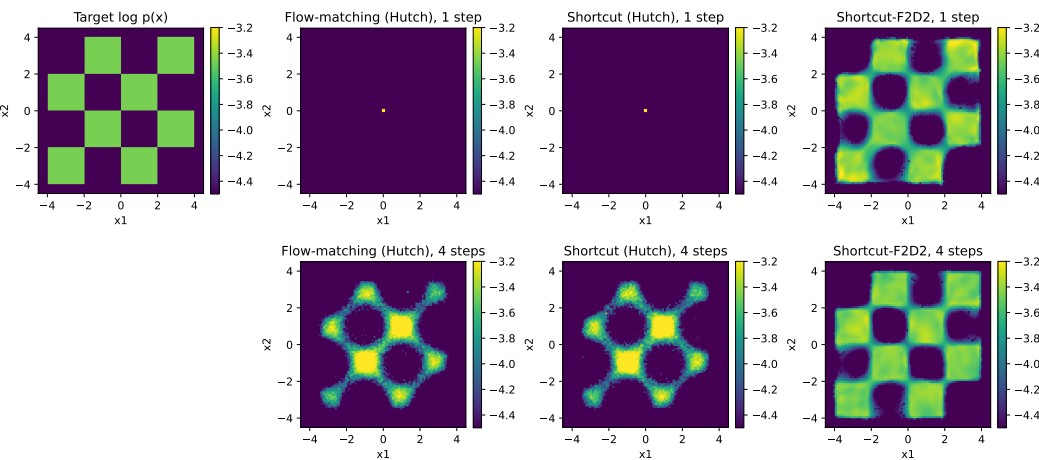

Figure 5: Log-likelihood comparison on 2D checkerboard generated samples among different models using forward integration.

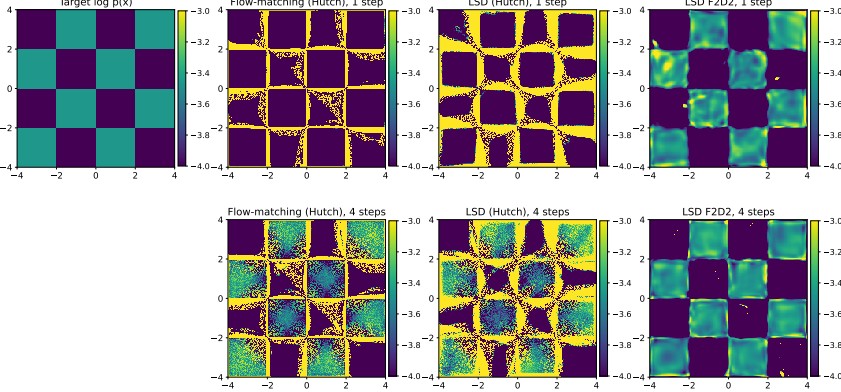

Figure 6: Log-likelihood comparison on 2D checkerboard ground truth dataset among different models using backward integration.

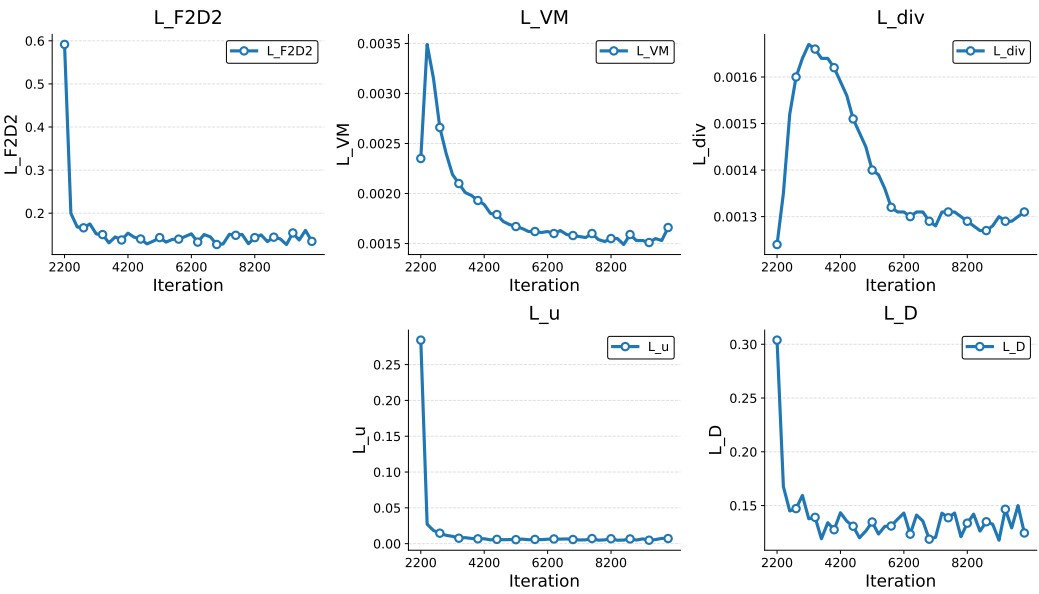

Figure 7: The Shortcut-Distill-F2D2 loss curve on CIFAR10.

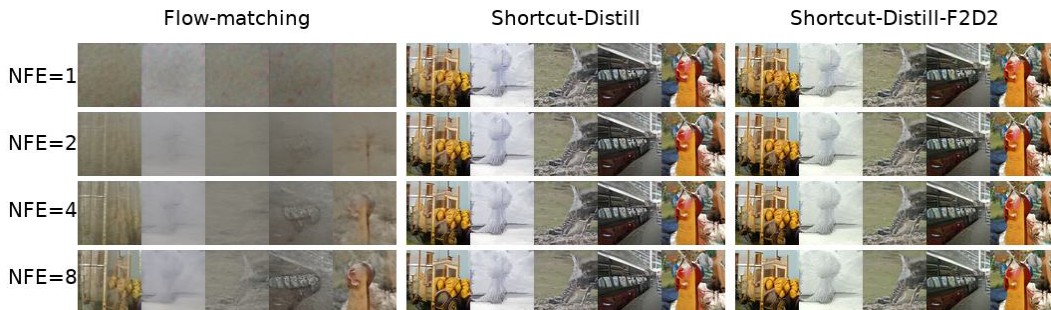

Figure 8: Imagenet $64 \times 64$ unconditional generation.

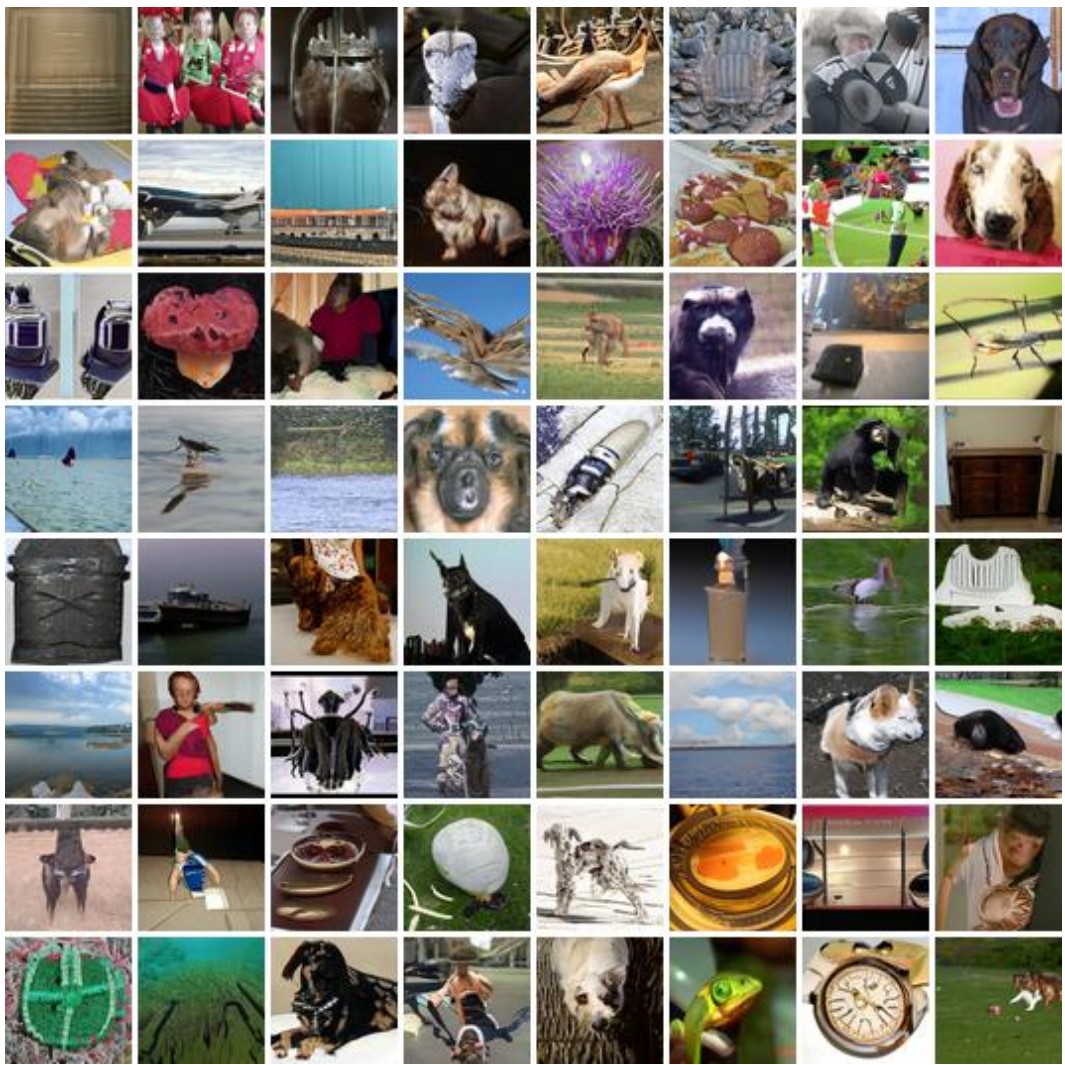

Figure 9: 8-step unconditional ImageNet $64 \times 64$ generation with our Shortcut-Distill.

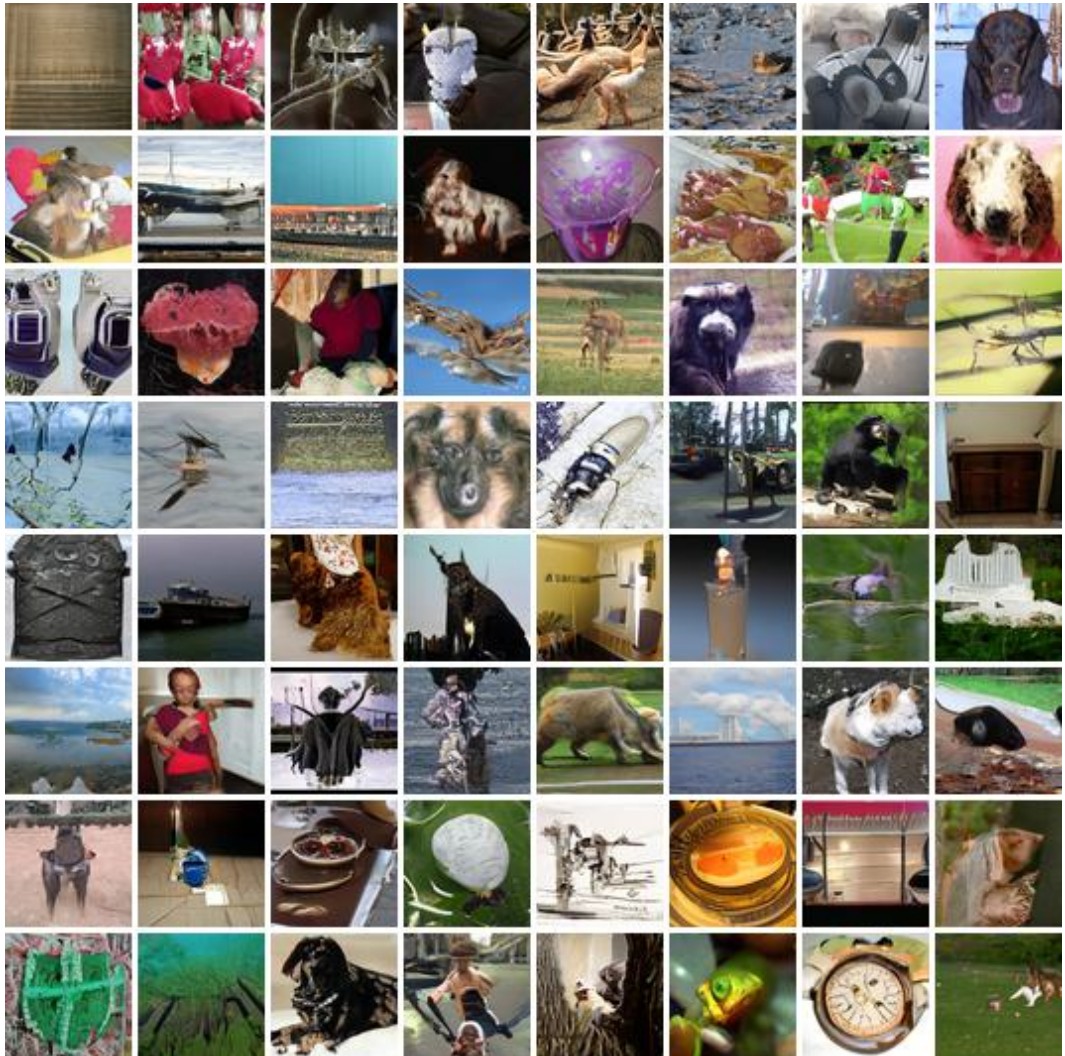

Figure 10: 2-step unconditional ImageNet $64 \times 64$ generation with our Shortcut-Distill.

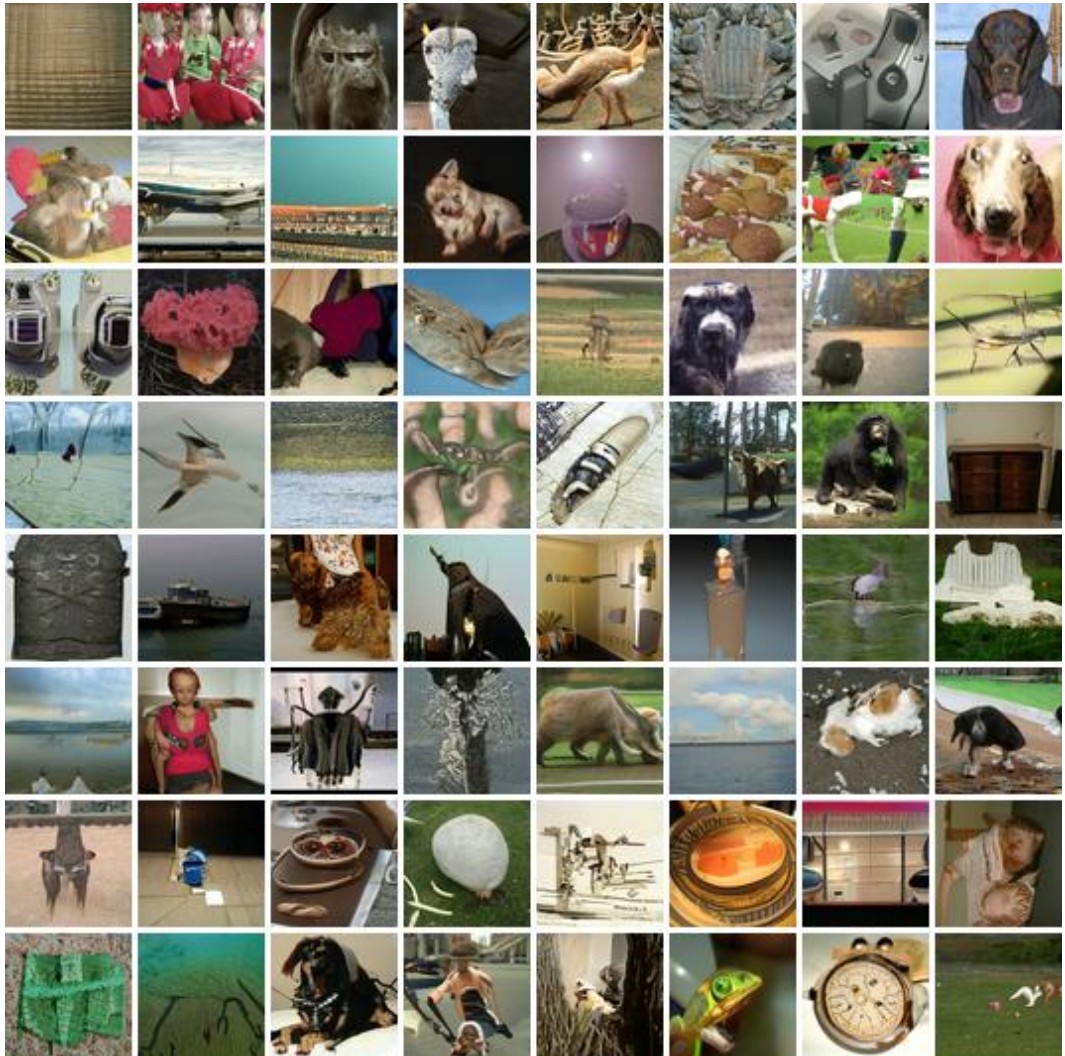

Figure 11: 8-step unconditional ImageNet $64 \times 64$ generation with our Shortcut-Distill-F2D2.

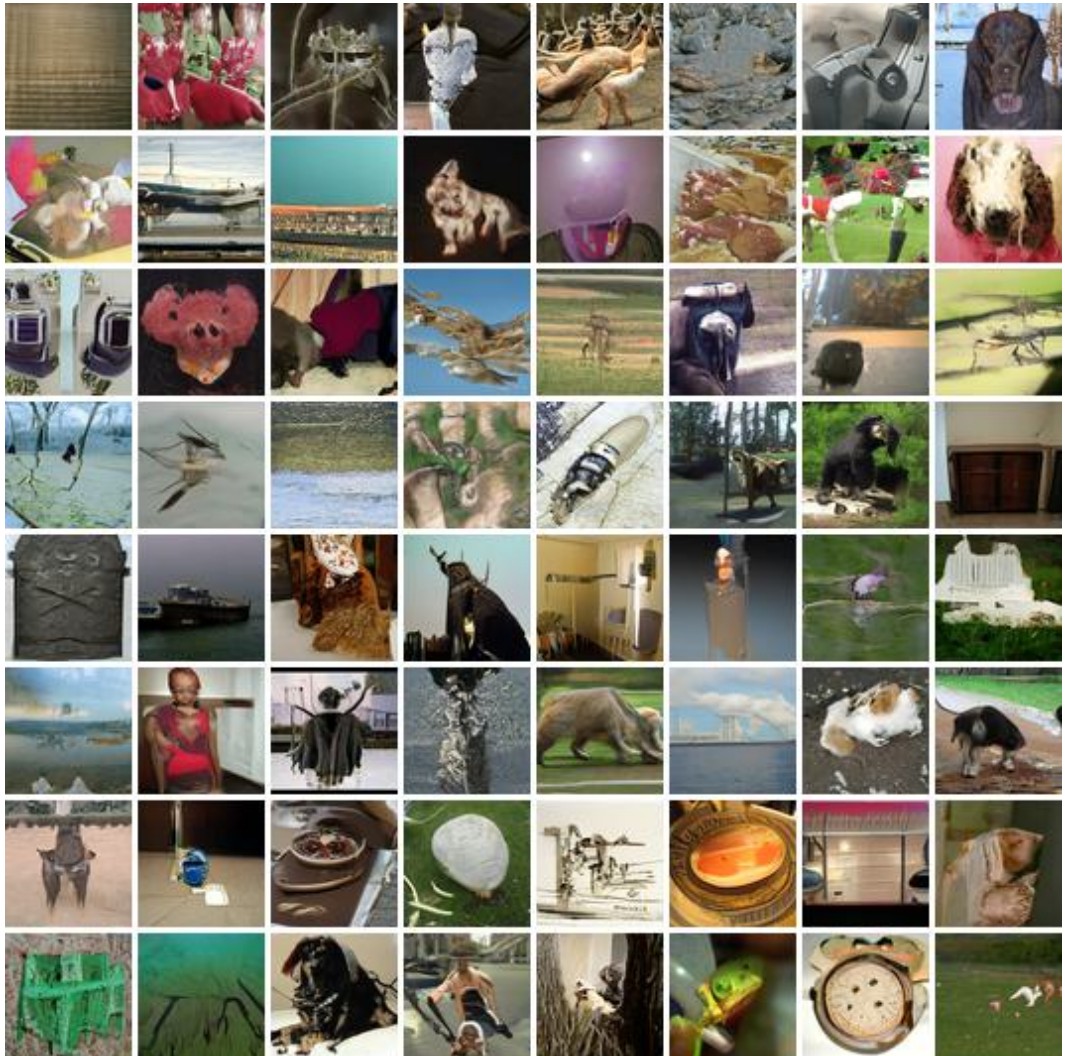

Figure 12: 2-step unconditional ImageNet $64 \times 64$ generation with our Shortcut-Distill-F2D2.

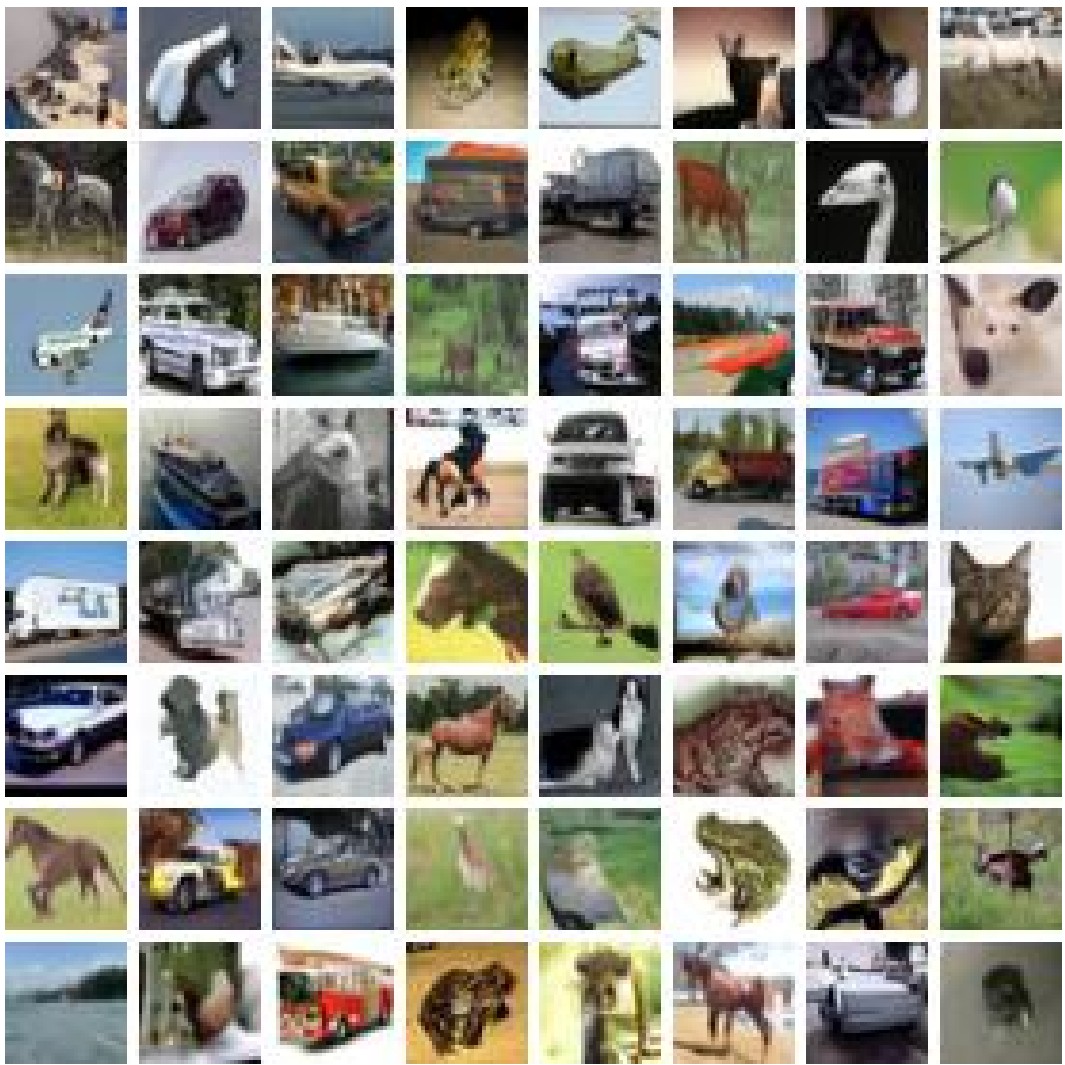

Figure 13: 8-step unconditional CIFAR-10 generation with our Shortcut-Distill.

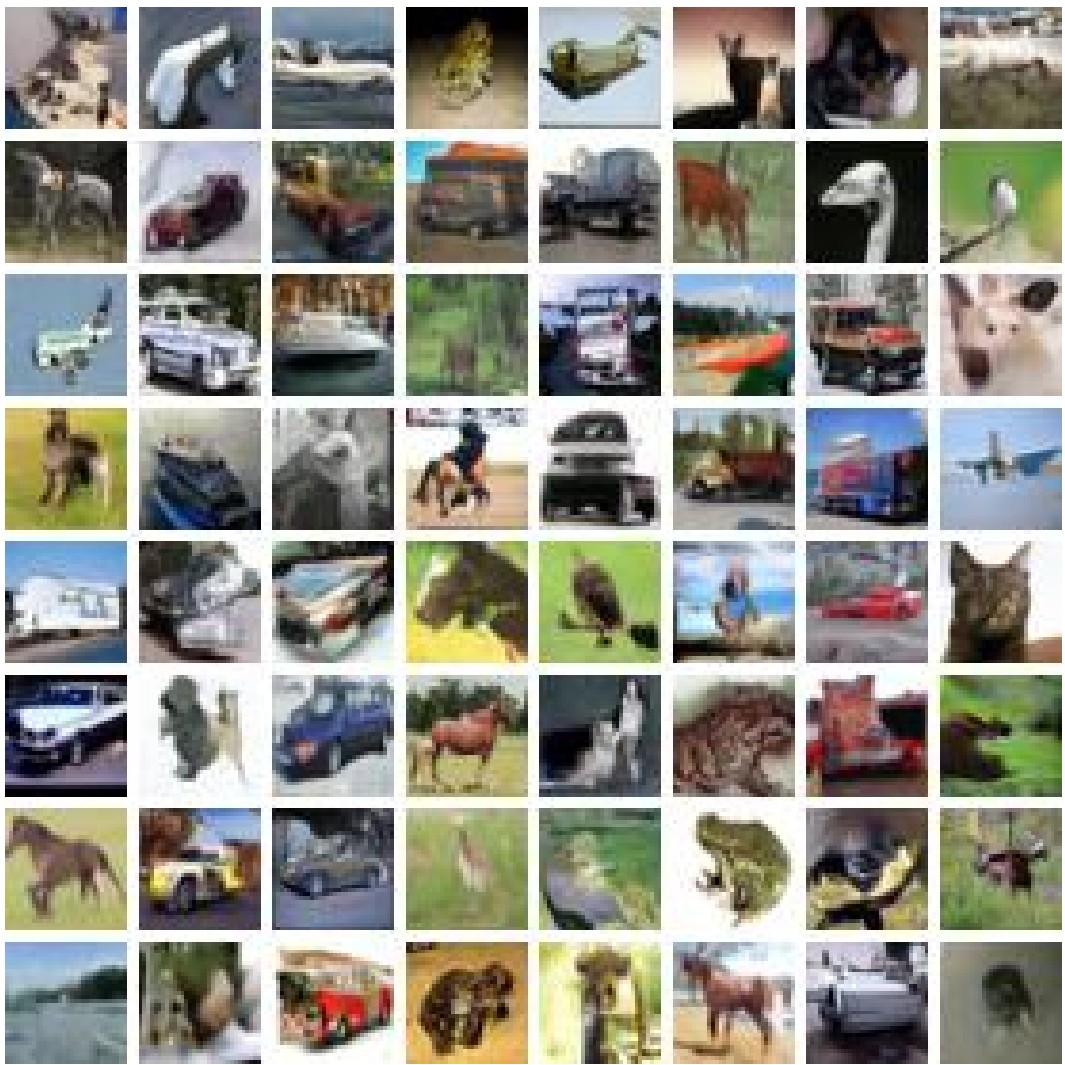

Figure 14: 2-step unconditional CIFAR-10 generation with our Shortcut-Distill.

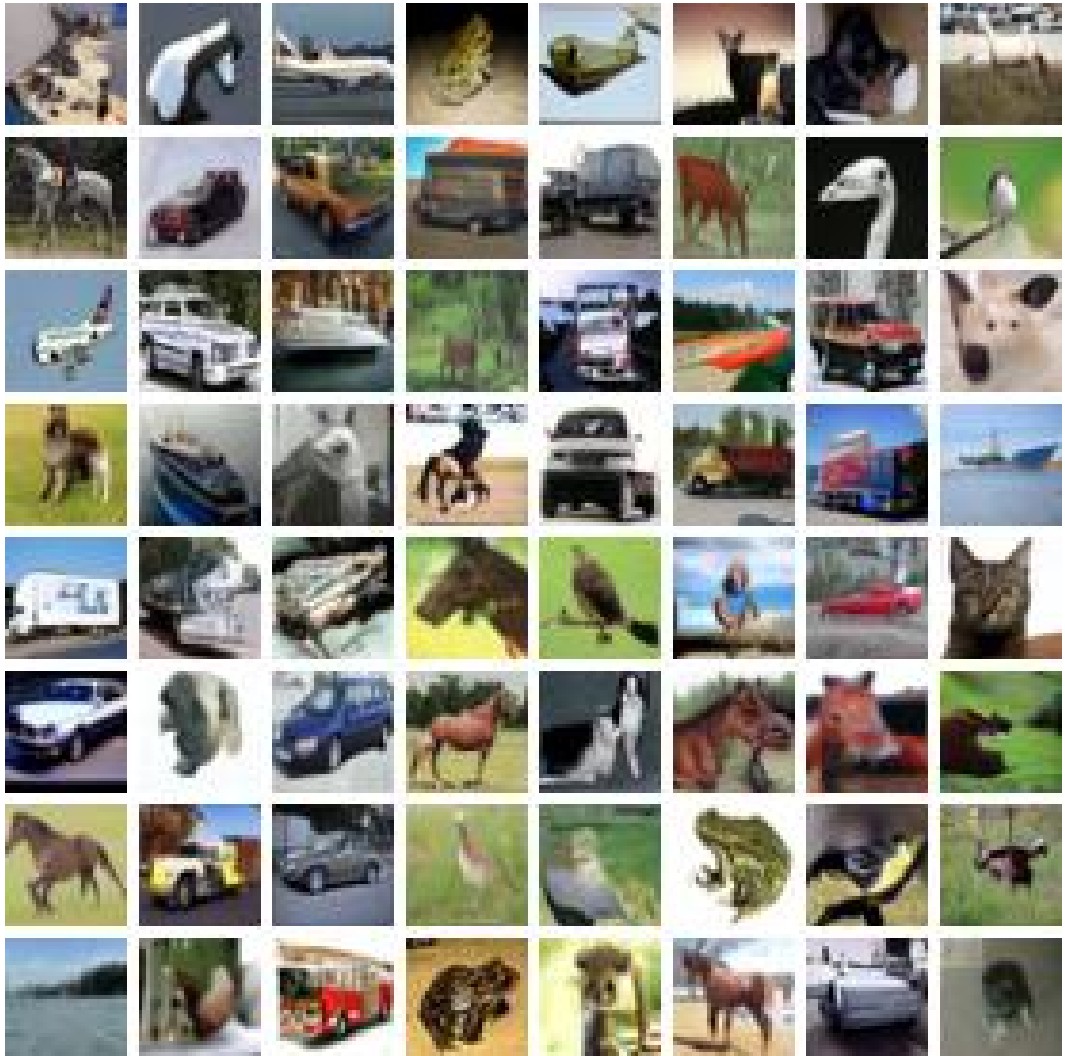

Figure 15: 8-step unconditional CIFAR-10 generation with our Shortcut-Distill-F2D2.

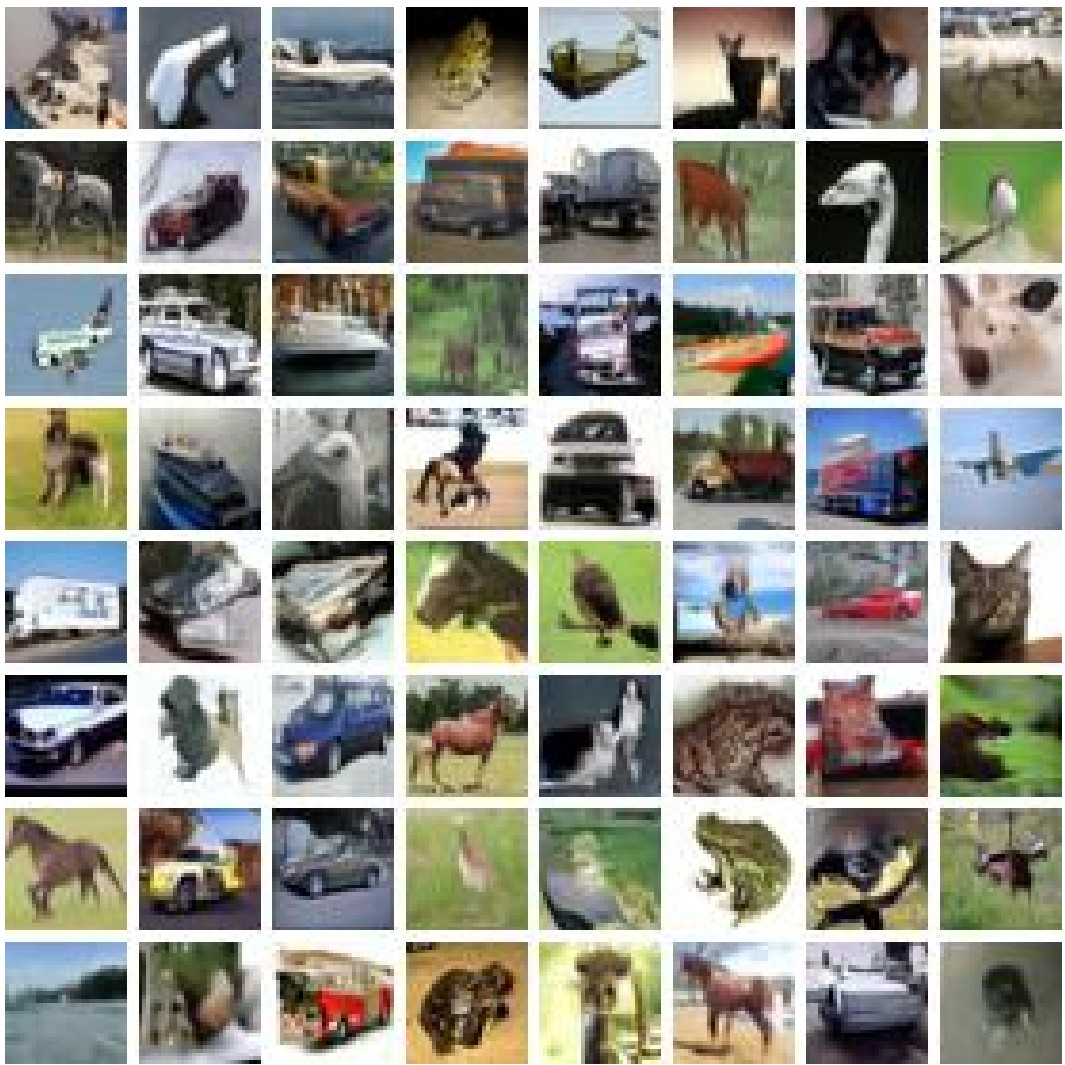

Figure 16: 2-step unconditional CIFAR-10 generation with our Shortcut-Distill-F2D2.

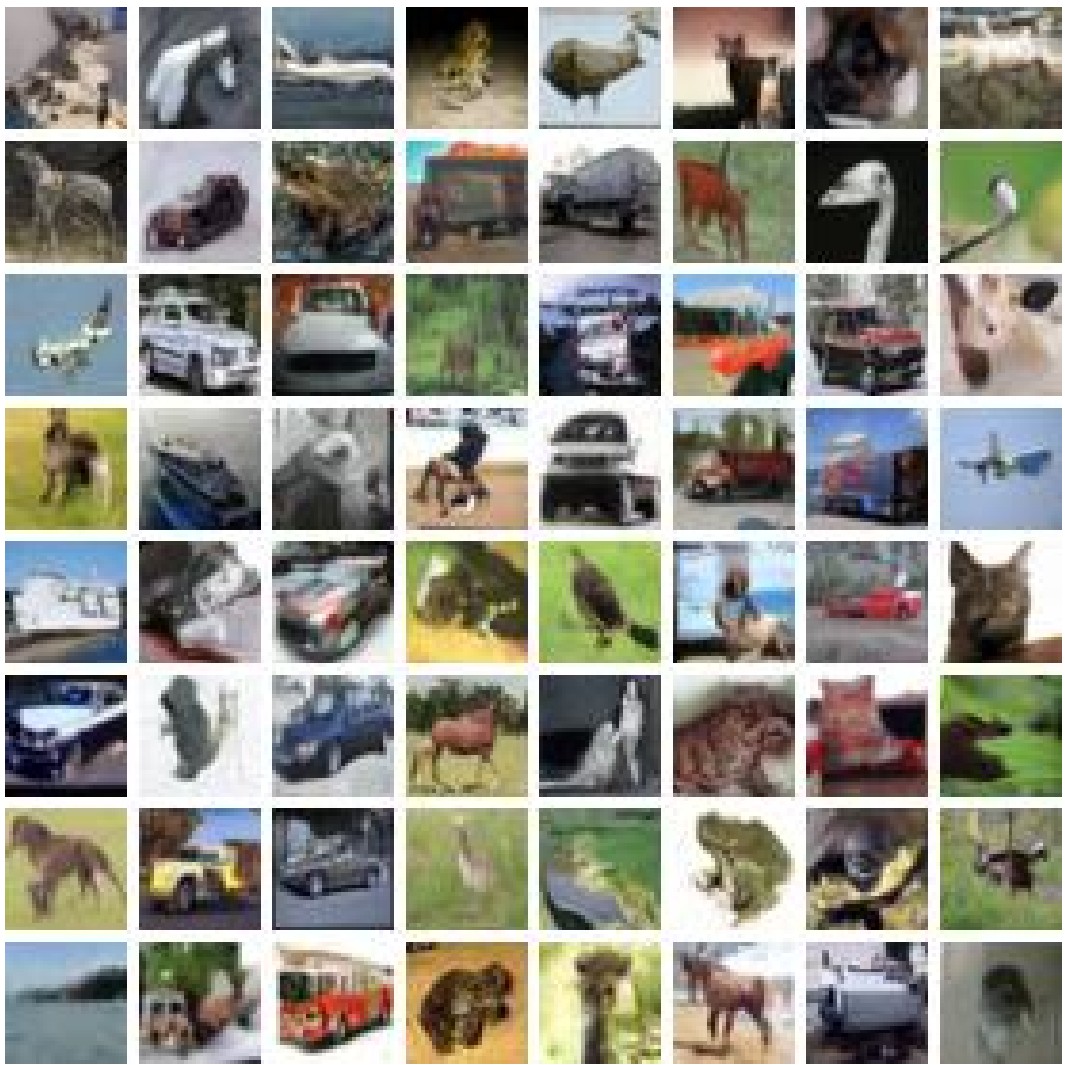

Figure 17: 8-step unconditional CIFAR-10 generation with our Shortcut-F2D2.

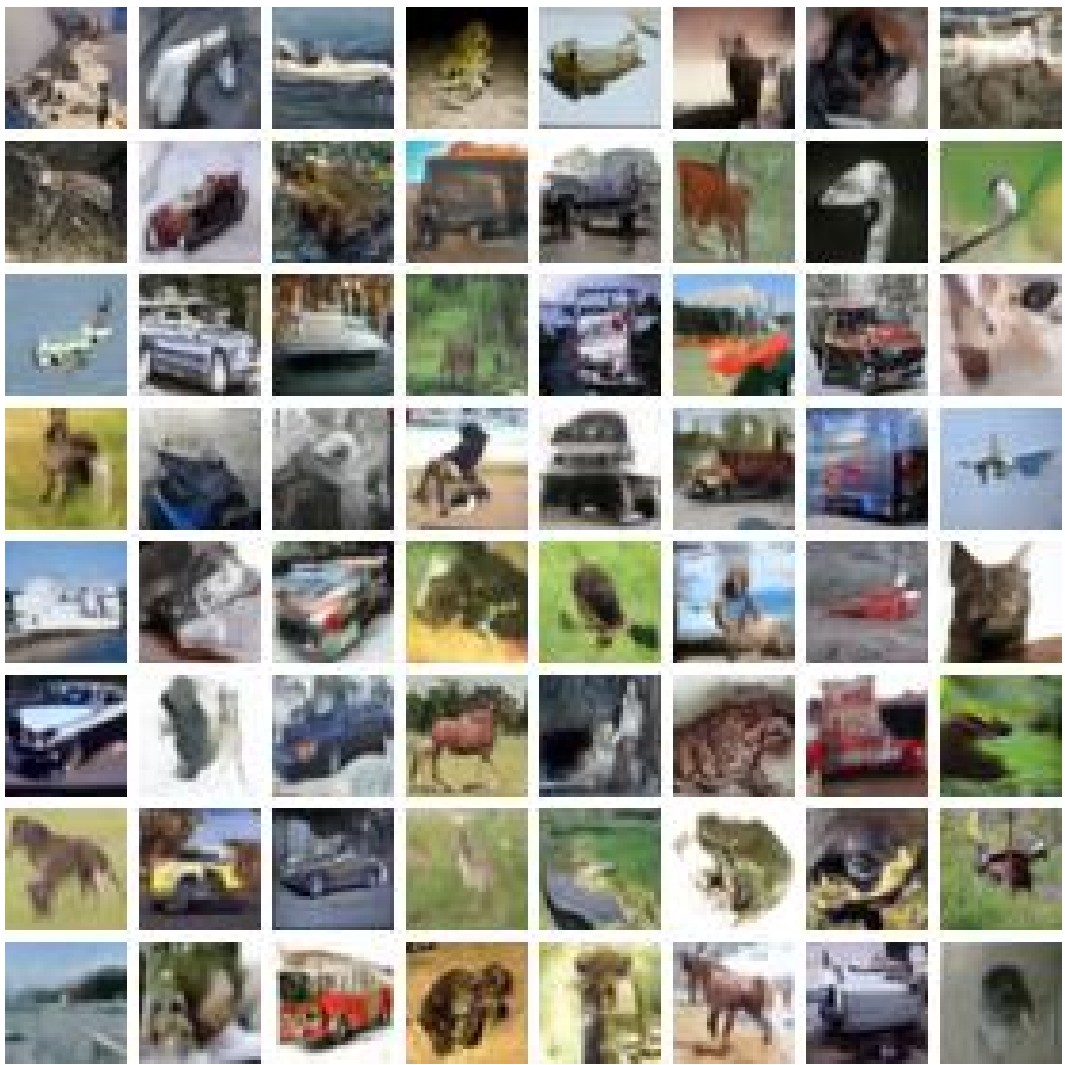

Figure 18: 2-step unconditional CIFAR-10 generation with our Shortcut-F2D2.

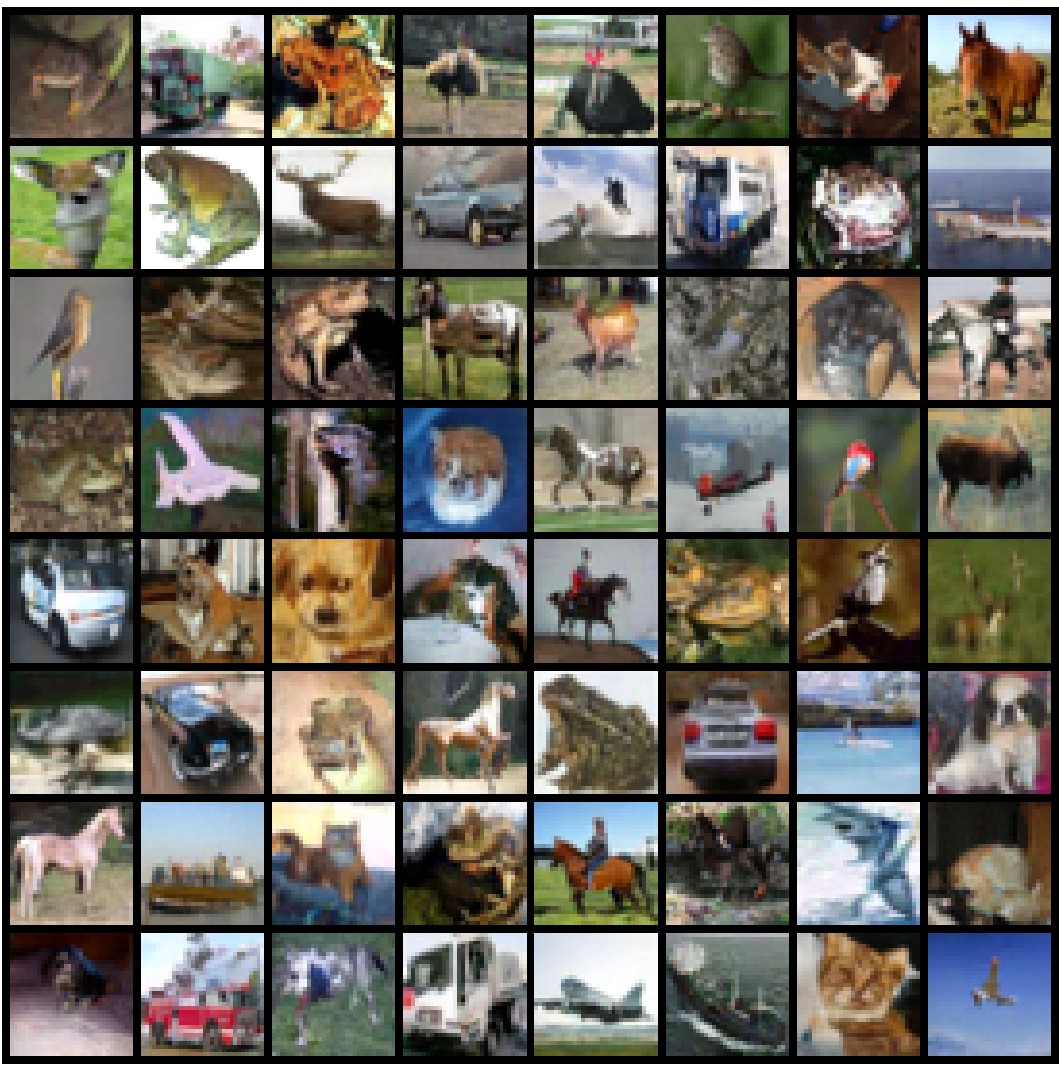

Figure 19: 2-step unconditional CIFAR-10 generation with MeanFlow-F2D2 (Ours).

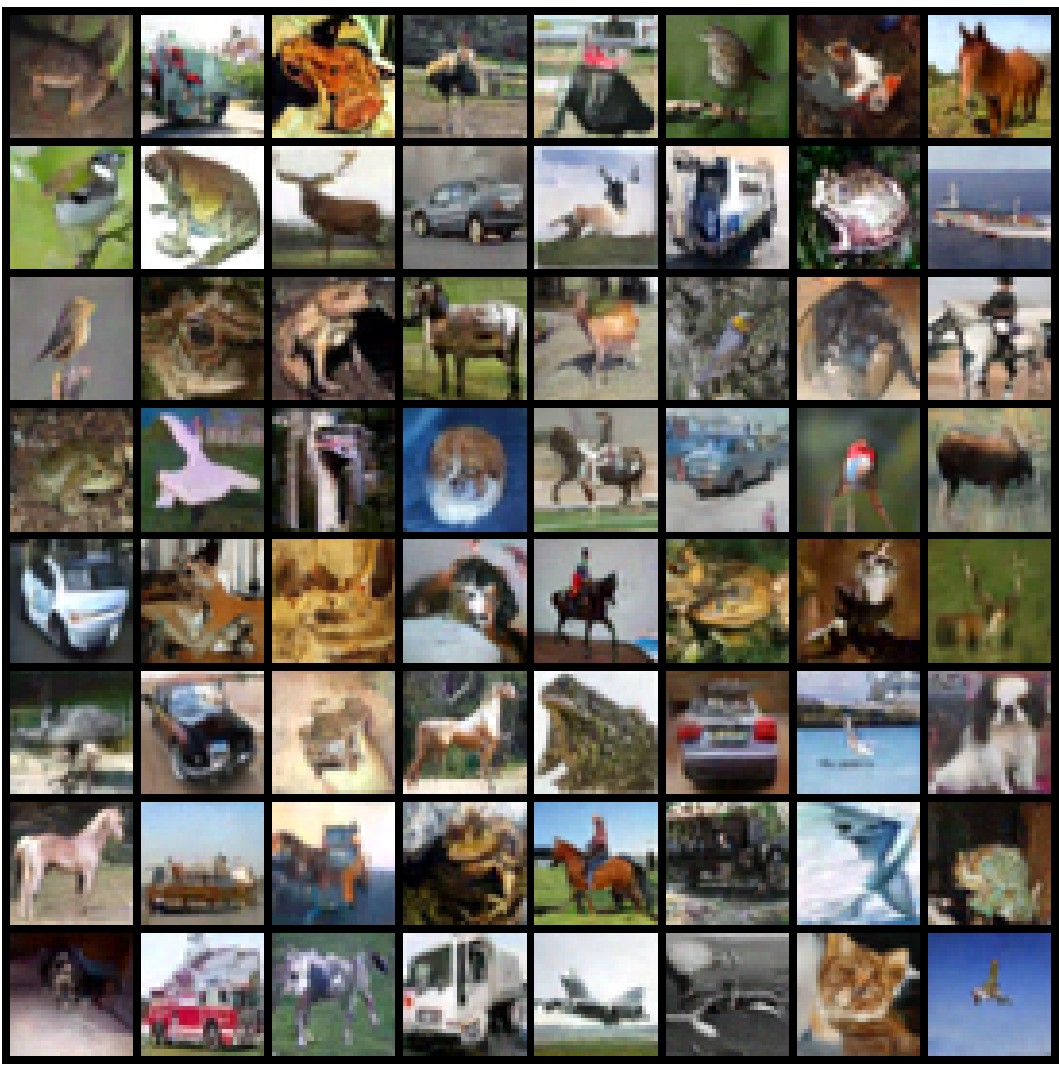

Figure 20: 1-step unconditional CIFAR-10 generation with MeanFlow-F2D2 (Ours).

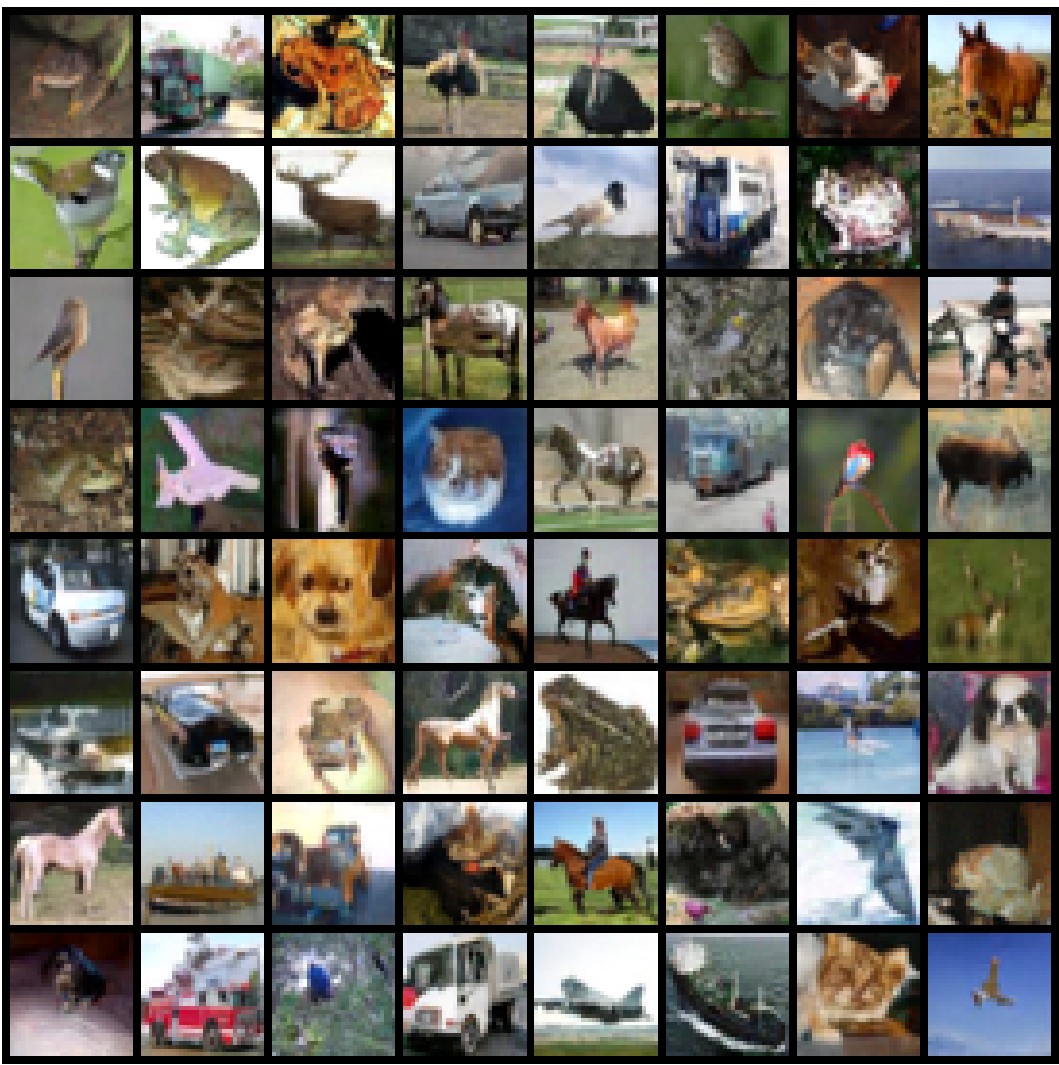

Figure 21: 2-step unconditional CIFAR-10 generation with our MeanFlow-F2D2-Self-Guidance.

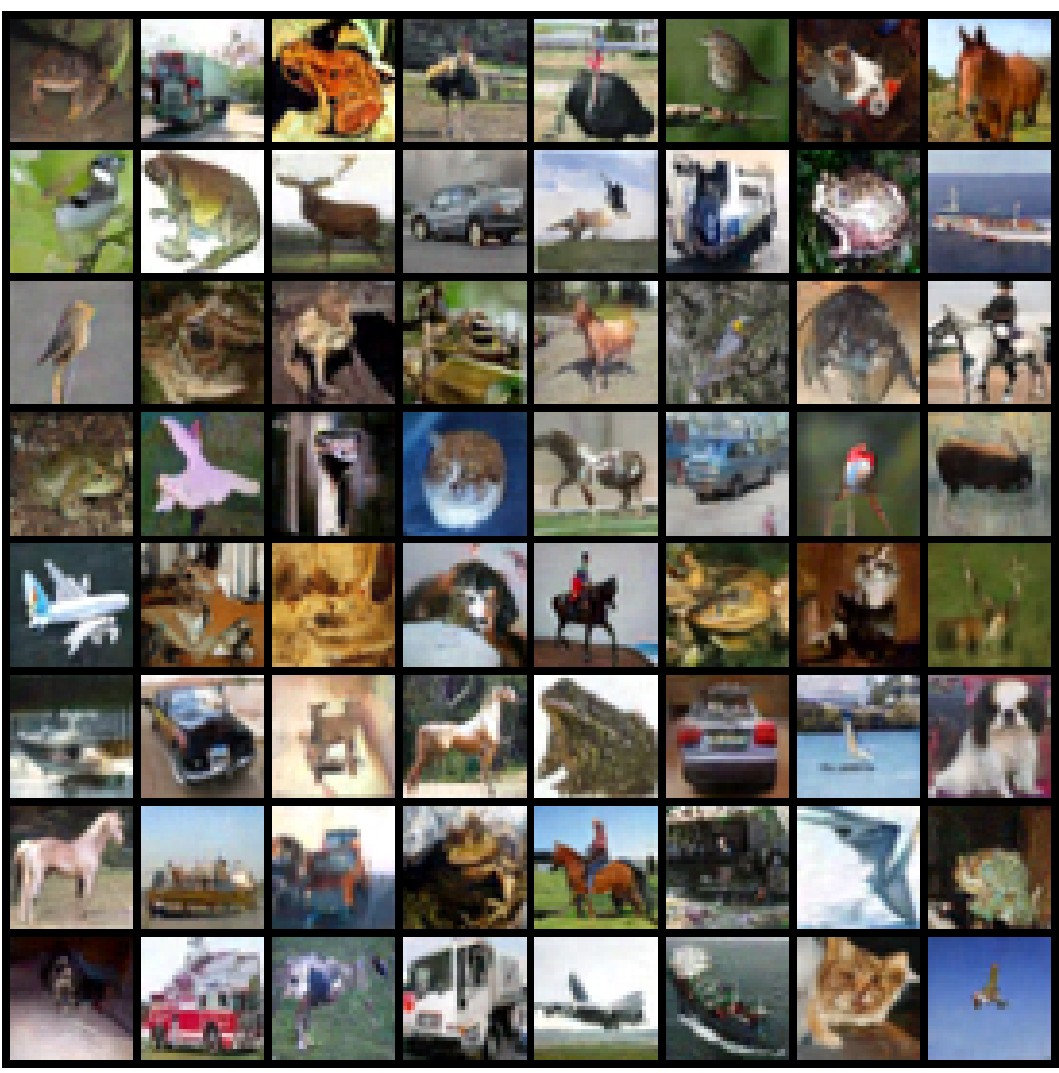

Figure 22: 1-step unconditional CIFAR-10 generation with our MeanFlow-F2D2-Self-Guidance.

## F  LIMITATIONS AND FUTURE WORKS

In this section, we discuss the limitations and future works of our method. First, F2D2 training requires careful early stopping: once the calibrated BPD value is reached, further training can potentially lead to overfitting or degraded likelihood estimation. Future work could address this through improved network architecture and auxiliary regularization. Second, on the ImageNet-$64 \times 64$ dataset, due to computational resources constraints, we only train with reduced model size and insufficient training iterations rather than an ideal large-scale configuration. We expect training with longer duration and larger models can further improve the performance. We also restrict ourselves to unconditional generation, which is substantially more challenging than conditional setups. Finally, the performance of our method is sensitive to divergence target scaling and its practical effectiveness in other architectures or modalities remains to be fully validated in future works.

## G  THE USE OF LARGE LANGUAGE MODELS (LLMS)

We use Large Language Models (LLMs) to refine and polish the manuscript. LLMs also support our code debugging, but they are not involved in developing the algorithms or conducting the experiments. The authors take full responsibility for the content of manuscript.

