# OpenReview forum: "Joint Distillation for Fast Likelihood Evaluation and Sampling in Flow-based Models"
_ICLR.cc/2026/Conference — ICLR 2026 Poster_

### Official Review · Reviewer_XjHG · 2025-10-27

**Soundness:** 2
**Presentation:** 3
**Contribution:** 3
**Rating:** 6
**Confidence:** 3

**Summary:**

The paper proposes Fast Flow Joint Distillation (F2D2), a framework that jointly distills (i) a few-step sampling flow map and (ii) a few-step log-likelihood evaluator for flow-matching models by learning a shared backbone with two heads: an average-velocity head and a cumulative-divergence head. The key idea is that sampling and likelihood evaluation requires solving a coupled ODE system that's driven by the same velocity field, making training a joint backbone neural network possible. The method is evaluated empirically with two families of few-step samplers: Shortcut-F2D2 and MeanFlow-F2D2, plus a practical Shortcut-Distill-F2D2 variant that warms up from a teacher flow. On CIFAR-10 and ImageNet-64, F2D2 is able to produce valid NLLs in a few NFEs and has competitive FID. Paper also proposes a simple maximum-likelihood self-guidance procedure that improves sample quality requiring an additional forward and backward pass.

**Strengths:**

- Framing sampling and likelihood evaluation within the flow-map framework for distillation using a shared backbone + two heads to reduce NFE is well-motivated.
- F2D2 is shown with semigroup Shortcut and Eulerian MeanFlow variants, demonstrating that F2D2 can plug in to different flow-map methods.
- Writing is clear and easy to follow.
- Hutchinson trace estimator for divergence, and a staged warm-start are sensible training choices.
- The maximum-likelihood self-guidance algorithm is simple and provides insight into inference-time scaling for image generation.

**Weaknesses:**

- The experiment section says F2D2 yields NLLs “close to the teacher’s BPD,” but several entries (e.g., MeanFlow-F2D2 at 2,8 steps) show BPD values materially below the teacher on CIFAR-10. What's more, the FID for MeanFlow-F2D2 is the best among settings. This raises the question of whether or not training on sampling and training on likelihood estimation are mutually beneficial training objectives, and whether or not the approach of using a shared backbone is justified.
- Using a 1024-step teacher as the “reference” BPD does not certify correctness of the few-step NLLs, it only checks consistency with a particular numerical property. An ablation on toy cases where the true likelihood are easily computed and comparing the F2D2 likelihood with the ground truth likelihood would better establish accuracy.
- Results are on CIFAR-10 and ImageNet-64 with unconditional generation. Users often cares more about higher resolutions and conditional tasks. The limitations section acknowledges compute constraints, but this still leaves open how the approach scales to large-scale, conditional generation.
- The paper argues for parameter sharing with dual heads. To validate the claim, it would be informative to see ablations on: (i) shared vs separate backbones, (ii) the dynamics of each loss term.

**Questions:**

1. Can you provide ablations on parameter sharing vs separate backbones and provide teacher quality for Shortcut-Distill-F2D2? What is the variance of the Hutchinson-based divergence estimator and how does this affect NLL stability and bias?
2. The experiment results show that MeanFlow-F2D2 has better FID than MeanFlow at low resolutions. MeanFlow conducted experiments on ImageNet 256x256 and the image quality is better. How does F2D2 perform at higher resolutions and will there be any qualitative failure modes when moving beyond 64×64? You mentioned early-stopping sensitivity for training in the Appendix, and this could raise some questions about scaling up training compute.
3. For Algorithm 2, can you provide experiment results about extending self-guidance beyond one iteration (e.g., x-axis being number of iterations and y-axis being FID/NLL)? Will there be any collapse after one iteration?
4. What is the total wall-clock time and compute resources used during training the divergence head compared to sampler and the teacher?

---

> ### Author Response · Authors · 2025-11-21
> **Author Rebuttal**
>
> We thank the reviewer for their constructive feedback and their recognition of our work. We would like to address their questions and concerns below.
>
> 1. **Whether the training objectives are mutually beneficial and justification on shared backbones:** Thank you for raising this concern. We would like to clarify that, the core contribution of our paper is enabling a **single model** to perform both fast sampling and fast likelihood evaluation **simultaneously**. Training separate backbones would simply create two independent models, defeating our main goal of creating a unified model for few-step generation and likelihood evaluation at the same time. Moreover, we would like to kindly point out that, even when F2D2's likelihood estimates are not perfectly calibrated, they remain dramatically more accurate than baseline methods, which produce entirely invalid predictions.
>
>     Regarding the mutual benefit of joint training: we do observe that joint optimization can influence FID in either direction. Therefore, we do not wish to claim F2D2 should be used solely to improve FID, and we have revised the introduction to clarify this. Rather, F2D2's primary contribution is to provide tractable likelihood evaluation via modular fintuning, with the additional benefit that accurate likelihoods enable test-time scaling methods like maximum likelihood self-guidance, which can then further improve FID (Figure 2). The fact that MeanFlow-F2D2 achieves both better FID and better likelihood in 3 out of 4 settings demonstrates that joint training can provide complementary signals, though this is not guaranteed across all configurations.
>
> 2. **Experiment with ground truth likelihood:** Thank you for this suggestion. We agree that experiments featuring likelihood evaluation with analytically tractable ground truth can significantly strengthen our paper. As a result, we have added Section 5.4 with 2D checkerboard experiments and compared our method against flow matching and vanilla shortcut model. As we can observe in Figure 3, flow matching and shortcut model completely fail at performing few-step likelihood evaluations while our method can achieve accurate results even with 1 NFE.
>
> 3. **Conditional generation:** Thank you for raising this point. We agree that conditional generation is generally a more popular task for ImageNet-scale and above datasets. However, we intentionally chose unconditional generation because it provides a clear evaluation for our main task: accelerating likelihood estimation and sample quality at the same time. Conditional NLL is ill-posted due to various confounding factors (e.g. CFG tricks and ambiguous marginal v.s. conditional target), and is not a standard practice in the likelihood evaluation literature.
>
> 4. **Variance of Hutchinson estimator and training stability:** Thank you for raising this detailed practical concern. With variance around 1.8e6 each time step, the Hutchinson estimator does produce values that are significantly larger in magnitude than the sampling range. Therefore, as we have mentioned in the Appendix C, we follow common deep learning practices and apply a 5e-5 scaling factor to the divergence target in order for the neural network to better learn the prediction. Following the reviewer’s suggestion, we have added an ablation study on the divergence scaling factor to validate this choice of hyperparameter in Appendix D Table 5 and our training loss curve in Appendix E Figure 5 to showcase the training stability. As we can observe, appropriate scaling enables simultaneous calibrated likelihood prediction and high sample quality, while scaling too strong can produce invalid NLL and scaling too weak can degrade overall performance.
>
> | Divergence Scaling Factor | 8 Steps |       | 4 Steps |       | 2 Steps |       | 1 Step |       |
> |:-------:|:-------:|:-----:|:-------:|:-----:|:-------:|:-----:|:------:|:-----:|
> |         |   NLL   |  FID  |   NLL   |  FID  |   NLL   |  FID  |   NLL  |  FID  |
> |   5e-4  |   3.27  |  5.16 |   2.43  |  6.31 |   1.26  |  7.06 |  -2.92 | 13.16 |
> |   5e-5  |   3.12  |  5.68 |   2.87  |  5.96 |   2.38  |  7.35 |  1.62  | 13.76 |
> |   5e-6  |   5.38  | 15.57 |   5.44  | 25.20 |   5.50  | 10.96 |  5.60  | 23.18 |
>
> In addition, we are currently compiling the results to show the training dynamics of each loss term and we will provide an update here as well as in the paper when they are available.

---

> > ### Author Response · Authors · 2025-11-21
> > **Author Rebuttal (cont')**
> >
> > 5. **Scaling beyond 64x64 resolution:** We appreciate this important question about the scalability of our method. Due to computational resource constraints, our current experiments are limited to 64x64 resolution or below. However, F2D2 is a model architecture agnostic framework and the theoretical foundation we present in this paper is also scaling independent. Our current experiments serve as proof of concept to demonstrate broad applicability that we expect industry-scale labs to adopt with greater computational resources.
> >
> > 6. **Multi-step self-guidance:** Thank you for this suggestion. We have implemented the self-guidance sampling algorithm with multi-step Adam optimization and present the results in Appendix D Figure 4. As we can observe from Figure 4, while adding optimization steps does not further improve the FID, the performance also does not collapse. We are interested in other variants of our self-guidance algorithm to better improve the performance as exciting future work directions.
> >
> > 7. **Computational cost & run time:** We thank the reviewer for bringing up this point. We conduct all our training on an 8-GPU L40S node. To train a typical F2D2 CIFAR10 model, it takes around 1 day for the flow matching teacher, 1 day for the vanilla shortcut or shortcut-distill baselines, and 5 hours for the F2D2 finetuning on top of it. For ImageNet64x64 models, it takes around 2 days for the flow matching teacher, 2 days for the vanilla shortcut or shortcut-distill model, and 5 hours for the F2D2 finetuning. For inference run time, we provide the wall-clock time to generate **50k** samples for each dataset using different numbers of NFEs in the table below. We have also added the computational cost and run time information in our revised Appendix C.
> >
> > | Dataset        | Hardware    | 8 NFE                 | 4 NFE                 | 2 NFE                 | 1 NFE                 |
> > |----------------|-------------|-----------------------|-----------------------|-----------------------|-----------------------|
> > | CIFAR-10       | 4× L40S     | 146 s (2.92 ms/img) | 83 s (1.66 ms/img) | 51 s (1.02 ms/img) | 40 s (0.80 ms/img) |
> > | ImageNet-64×64 | 8× L40S     | 289 s (5.78 ms/img) | 160 s (3.20 ms/img) | 95 s (1.90 ms/img)  | 63 s (1.26 ms/img)  |

---

> ### Comment · Reviewer_XjHG · 2025-11-27
>
> Thank you for addressing the concerns. I'll keep my current ratings and monitor ongoing discussions.

---

> > ### Author Response · Authors · 2025-11-27
> > **Updates on the training dynamics of individual loss terms**
> >
> > Thank you so much for your response! We would like to provide an additional update regarding the demonstration of the training dynamics of individual loss terms in our F2D2 training as per requested in the original review. In our updated Figure 5 in Appendix E, we show the loss curves of all loss terms and the combined total loss for our Shortcut-Distill-F2D2 training on CIFAR10, and we can observe stable training with expected small fluctuations native to uniformly
> > random timestep selection at each iteration for all loss terms.
> >
> > We hope this further addresses the reviewer's concerns and we are happy to answer any other questions that the reviewer may have. Thank you again for your time and efforts!

---

### Official Review · Reviewer_RL2w · 2025-10-30

**Soundness:** 3
**Presentation:** 3
**Contribution:** 2
**Rating:** 2
**Confidence:** 4

**Summary:**

This paper targets an important problem of making FM based generative models to have fast likelihood estimation.  The proposed method is based on flow map distillation methods in the sampling space, such as short-cut model and meanflow. The authors conduct experiments on CIFAR-10 and ImageNet-64 to demonstrate that their NLL can be effectively calculated in a single step.

**Strengths:**

- This paper focuses on an important problem and is well motivated, as the author mentioned, fast likelihood computation could open up a wide area of future work, such as in the RL community.

- The writing is clear and easy to follow.
- The proposed method is simple and straightforward without the need of adversarial training or other additional tricks.

- The result in CIFAR-10 demonstrated that the proposed model can achieve state-of-the-art sampling quality in one step while having good NLL performance.

**Weaknesses:**

While the proposed approach sounds promising, the experiment results are lacking to fully support the paper's claim.

- The ImageNet 64x64 model's FID is far from state-of-the-art.
- MF has been reportedly by the community being not stable, adding more loss term could make it more stable for larger dataset/model. The scalability of the proposed approach is not studied.
- One important baseline Tarflow[1] is missing given it can also do fast NLL calculation and sampling.

[1] Zhai, S., et al. "Normalizing flows are capable generative models (2025)." URL https://arxiv. org/abs/2412.06329.

**Questions:**

- Why did the FID drop in Table 1 for meanflow with F2D2 compared to without ?  Limited network capabilities?
- Could the author clarify why the meanflow-F2D2 is initiated from a short-cut-Distill model ?  This makes it harder to judge the effectiveness of meanflow to your likelihood distillation.

---

> ### Author Response · Authors · 2025-11-21
> **Author Rebuttal**
>
> We thank the reviewer for their thoughtful comments. We appreciate their positive assessment of our method and exposition, and would like to address their concerns below.
>
> 1. **ImageNet-64x64 FID**: Thank you for raising this point. We would like to clarify that our ImageNet-64x64 experiment focuses on **unconditional** generation, whereas SOTA results on this dataset are achieved with **class-conditional** modeling. Conditional generation is substantially easier, and the SOTA models typically use significantly larger models with far more compute (e.g., TarFlow requires 450M+ parameters and 2 weeks of training v.s. our model has 200M parameters with 2 days of training). We chose unconditional generation because it provides a clear evaluation for our main task: accelerating likelihood estimation and sample quality at the same time. Conditional NLL is ill-posted due to various confounding factors (e.g. CFG tricks and ambiguous marginal v.s. conditional target), and is not a standard practice in the likelihood evaluation literature.
>
>     Our ImageNet-64x64 FID is close to SOTA for unconditional models at this computational scale. Our teacher model, which is trained with SOTA method flow matching, produces 23.39 FID with 200 step sampling. In the meantime, our 8-step Shortcut-Distill-F2D2 achieves 29.49 FID while enabling calibrated NLL estimation, demonstrating the effectiveness of F2D2 at this scale.
>
>     Finally, we would like to emphasize that the main contribution of our paper is methodological rather than attaining SOTA with specific architectures. F2D2 provides a general framework applicable to any flow-map-based method, and enables simultaneous fast sampling and fast likelihood evaluation for the first time for diffusion/flow matching-based models. Our experiments serve as proof of concept to demonstrate broad applicability that we expect industry-scale labs to adopt with greater computational resources.
>
> 2. **Stability of MeanFlow:** We understand the concern that MeanFlow requires a number of subtle design decisions to implement; however, when implemented correctly, it does achieve strong results as demonstrated in our Table 1 and in the original MeanFlow paper. More importantly, MF is just one of many instantiations of our approach – F2D2 applies to any flow map method. If stability challenges arise with MeanFlow, our framework works with other base methods like shortcut models.
>
> 3. **Comparison with TarFlow:** Thank you for the pointer. TarFlow represents a complementary approach to fast likelihood evaluation, but differs fundamentally from our work in both methodology and scope.
>
>     (a) Methodologically, TarFlow is a normalizing flow model and it achieves direct likelihood evaluation through its architectural design. On the other hand, our F2D2 is a general and modular distillation framework that adds fast likelihood capability to any existing few-step flow map model while maintaining high sample quality, which is architecture agnostic.
>
>     (b) Scope wise, TarFlow requires 450M+ parameters and 2 weeks of training and reports 2 minutes for 32 samples on 1 A100 GPU (i.e. 256 images on 8 GPUs). Our model has 200M parameters with 2 days of training, and is able to generate 50k images in around 1 minute on 8 L40S GPUs. Even accounting for architectural differences, we estimate that F2D2 can still offer substantially faster inference (>100× estimated speedup), though rigorous comparison will require scaling our model to TarFlow level resources, which is beyond our current computational budget.
>
>     That being said, we do recognize the missed opportunity to discuss the related literature regarding normalizing flows, and therefore we have provided a detailed discussion of TarFlow and broader normalizing flow models in the related work section.
>
> 4. **”Why did FID drop for MeanFlow with F2D2”:** Thank you for mentioning this observation. We would like to first clarify that, FID dropping indicates improved sample quality, not degradation. In other words, MeanFlow-F2D2 achieves better FID than baseline MeanFlow in 3 of 4 settings. As a result, this improvement is a positive result demonstrating that our F2D2 can enable both likelihood evaluation and sample quality simultaneously
>
>     Why joint distillation improves FID for MeanFlow remains an interesting question for future study. We hypothesize that this can be attributed to the regularization effect from the multi-task training objective to the strong MeanFlow objective, though further investigation is needed to identify the precise causes.
>
> 5. **”Why is the MeanFlow-F2D2 initiated from a ShortCut-Distill model”:** We apologize for the confusion in our wording. MeanFlow-F2D2 is initialized from the official pre-trained MeanFlow checkpoint, not from Shortcut-Distill. The statement "we use the same scalar head for CIFAR-10 with Shortcut-Distill-F2D2" refers to using the same network architecture for the divergence head, not the same trained weights.

---

> ### Author Response · Authors · 2025-12-02
> **Updates on ImageNet-64x64 model results**
>
> Dear reviewer,
>
> Thank you again for your thoughtful feedback. During the course of the rebuttal period, we have been trying to scale up our ImageNet-64x64 experiment, and we would like to provide an update on it here.
>
> We have managed to scale up our model from around 200M to 324M, and we extended the training time from 5 days to two weeks in total. By doing so, we obtain the results in the table below (also shown in the updated Table 2).
>
> | Method                         | 8 Steps NLL | 8 Steps FID | 4 Steps NLL | 4 Steps FID | 2 Steps NLL | 2 Steps FID | 1 Step NLL | 1 Step FID |
> |--------------------------------|------------:|------------:|------------:|------------:|------------:|------------:|-----------:|-----------:|
> | Flow Matching                  |      -6.41  |       31.60 |     -15.87  |       68.55 |     -35.23  |      170.00 |    -74.54  |     363.39 |
> | Shortcut-Distill (Ours)        |      -9.03  |       19.47 |     -22.30  |       21.73 |     -49.01  |       28.12 |   -102.07  |      42.72 |
> | Shortcut-Distill-F2D2 (Ours)   |       3.51  |       21.91 |       3.94  |       24.05 |       3.97  |       29.83 |      1.54  |      44.02 |
>
>
> As we can observe, we are able to achieve FID scores comparable to TarFlow (18.42 v.s. 21.91) but with a model that is **1.5x smaller** in size and with **81-372x speedup**.
>
> We hope this result addresses reviewer's concerns regarding the ImageNet-64x64 performance of our model. Thank you again for your constructive comments and suggestions!

---

### Official Review · Reviewer_5Bkt · 2025-10-30

**Soundness:** 2
**Presentation:** 2
**Contribution:** 2
**Rating:** 4
**Confidence:** 3

**Summary:**

This paper introduces fast flow joint distillation(F2D2), which jointly learns sampling trajectory and divergence using a joint self-distillation  objective loss and a single model to enable few-step sampling and fast log-likelihood evaluation in flow-based models. It shows that the method is compatible with Shortcut and Meanflow Models and can be further distilled. The proposed method performs on image-based dataset and shows optimistic results.

**Strengths:**

1. It's interesting to learn vector field and divergence by sharing the same backbone with separate prediction heads since they are related to each other.
2. The proposed method can be extended to shortcut model and meanflow model.

**Weaknesses:**

1. The objective loss of shortcut F2D2 is composed of 4 components. It is not quite clear if any one of the components might dominate the whole training and make the performance better. An ablation study is required here. Also, are the 4 components equally weighted? It appears equally weighted from the equation but what if the weights are tuned.
2. Though the idea to learn velocity field and divergence is interesting, I am wondering if it might lead to instability during training since velocity field learns the directional vector and divergence learns the expectation of the trace through Hutchinson estimator, which involves a gradient of vector field. What if they do not share the same numerical range? Again, it is better to have an ablation study here.
3. For the experiments, no computation cost and runtime is provided.

**Questions:**

Following the weaknesses:

1. In table 1, I found it is hard to interpret the results as no method performs consistently well than the others across different sampling steps, either it is existing method or the proposed methods. The results from shortcut distilling method outperforms the shortcut model sometimes. I am wondering if there is actually no pattern and the good performance happens by chance.
2. In Shortcur-Distilll-F2D2, it only talks about the vector field training, how's divergence trained here?
3. How are invalid NLLs computed in table 1?
4. I am not pretty sure why we should learn the divergence term from the beginning. Presumably we can learn instantaneous/average vector fields well, and this will induce the marginal density distribution for each $\rho_t(x)$, then it will automatically the continuity function, which means we know the divergence in the meanwhile. If this is correct, learning divergence is redundant. Please correct me if I am wrong.

---

> ### Author Response · Authors · 2025-11-21
> **Author Rebuttal**
>
> We thank the reviewers for their thoughtful comments and suggestions. We would like to address their concerns below.
>
> 1. **Loss weighting:** Thank you for this question. The four loss components are theoretically motivated by the flow map characterization (Proposition 3.3): $L_{VM} + L_{u}$ enforce the sampling flow map conditions, while $L_{div} + L_{D} $ enforce the likelihood flow map conditions. In principle, these should be equally weighted since they jointly characterize a valid flow map. That being said, we agree with the reviewer that it would be interesting to see if additional empirical loss weighting can further improve the performance. We are actively  conducting the requested ablation study on individual loss components. However, due to our computational constraint, we are still waiting for the experiments to be completed. We will update results in the discussion as well as in the revised paper as soon as they are available.
>
> 2. **Variance of Hutchinson estimators:** Thank you for raising this detailed practical concern. With variance around 1.8e6 each time step, the Hutchinson estimator does produce values that are significantly larger in magnitude than the sampling range. Therefore, as we have mentioned in the Appendix C, we follow common deep learning practices and apply a 5e-5 scaling factor to the divergence target in order for the neural network to better learn the prediction. Following the reviewer’s suggestion, we have added an ablation study on the divergence scaling factor to validate this choice of hyperparameter in Appendix D Table 5 and our training loss curve in Appendix E Figure 5 to showcase the training stability. As we can observe, appropriate scaling enables simultaneous calibrated likelihood prediction and high sample quality, while scaling too strong can produce invalid NLL and scaling too weak can degrade overall performance.
>
> | Divergence Scaling Factor | 8 Steps |       | 4 Steps |       | 2 Steps |       | 1 Step |       |
> |:-------:|:-------:|:-----:|:-------:|:-----:|:-------:|:-----:|:------:|:-----:|
> |         |   NLL   |  FID  |   NLL   |  FID  |   NLL   |  FID  |   NLL  |  FID  |
> |   5e-4  |   3.27  |  5.16 |   2.43  |  6.31 |   1.26  |  7.06 |  -2.92 | 13.16 |
> |   5e-5  |   3.12  |  5.68 |   2.87  |  5.96 |   2.38  |  7.35 |  1.62  | 13.76 |
> |   5e-6  |   5.38  | 15.57 |   5.44  | 25.20 |   5.50  | 10.96 |  5.60  | 23.18 |
>
> 3. **Computational cost & run time:** We thank the reviewer for bringing up this point. We conduct all our training on an 8-GPU L40S node. To train a typical F2D2 CIFAR10 model, it takes around 1 day for the flow matching teacher, 1 day for the vanilla shortcut or shortcut-distill baselines, and 5 hours for the F2D2 finetuning on top of it. For ImageNet64x64 models, it takes around 2 days for the flow matching teacher, 2 days for the vanilla shortcut or shortcut-distill model, and 5 hours for the F2D2 finetuning. For inference run time, we provide the wall-clock time to generate **50k** samples for each dataset using different numbers of NFEs in the table below. We have also added the computational cost and run time information in our revised Appendix C.
>
> | Dataset        | Hardware    | 8 NFE                 | 4 NFE                 | 2 NFE                 | 1 NFE                 |
> |----------------|-------------|-----------------------|-----------------------|-----------------------|-----------------------|
> | CIFAR-10       | 4× L40S     | 146 s (2.92 ms/img) | 83 s (1.66 ms/img) | 51 s (1.02 ms/img) | 40 s (0.80 ms/img) |
> | ImageNet-64×64 | 8× L40S     | 289 s (5.78 ms/img) | 160 s (3.20 ms/img) | 95 s (1.90 ms/img)  | 63 s (1.26 ms/img)  |
>
> 4. **How to interpret Table 1:** We appreciate this clarifying question. The purpose of Table 1 is not to compare and determine which flow map method is the best, but rather to demonstrate that F2D2 is a general framework applicable to diverse flow-map based (self-)distillation methods. The key observation from Table 1 is that, regardless of the base method, F2D2 consistently transforms their invalid likelihood predictions into calibrated ones while maintaining their high sample quality. This systematic improvement occurs across the board and is not a coincidence.
>
> Regarding the improvement of Shortcut-Distill v.s. shortcut model, we propose Shortcut-Distill to better leverage the training signal from the teacher model to improve the FID performance of the shortcut model. However, it is orthogonal to the main contribution of F2D2 and hence can be combined to form Shortcut-Distill-F2D2, further demonstrating the modularity of F2D2 framework.

---

> > ### Author Response · Authors · 2025-11-21
> > **Author Rebuttal (cont')**
> >
> > 5. **Divergence training of Shortcut-Distill-F2D2:** The divergence training in Shortcut-Distill-F2D2 follows the same procedure as Shortcut-F2D2, using the joint distillation losses in Equations 3.8-3.9. The only difference between Shortcut-Distill-F2D2 and Shortcut-F2D2 is that Shortcut-Distill-F2D2 uses teacher velocity supervision instead of self-consistency for the sampling component, but this does not affect the divergence training.
> >
> > 6. **How are the invalid NLL computed & Why is learning an additional divergence flow map necessary:** We thank the reviewer for these questions as they help clarify a crucial point of our method. The reviewer's intuition is correct that flow maps can recover instantaneous velocity through their tangent condition $u_\theta(x_t,t,t) \approx v(x_t,t)$. This means flow map models trained only for sampling can, in principle, evaluate likelihood by computing $div(u_\theta(x_t,t,t))$ and solving for Eq 2.3 and 2.4 using numerical solvers like Euler. **This is precisely how we calculate the NLL values for the baselines (methods without F2D2) in Table 1.**
> >
> > However, as we can observe in Table 1, this approach fails catastrophically when using very few (<10) NFEs and renders invalid NLL predictions. (We mark NLL predictions that are less than 0 BPD, which implies that the model assigns greater than 1 probability to the 8-bit images and violates probability axioms.) This is why learning to directly distilling the integrated divergence into a flow map is necessary: F2D2 solves this problem by directly predicting the integrated divergence rather than integrating instantaneous estimates, which reduces the computational requirement of calculating accurate likelihood from 100-1000 NFEs to <10 NFEs. We appreciate the reviewer for asking this question and we have added this clarification to the revised implementation detail section.

---

> > > ### Author Response · Authors · 2025-11-27
> > > **Updates on the loss weighting ablation study**
> > >
> > > Dear reviewer,
> > >
> > > Thank you again for your constructive feedback! We would like to provide an ablation on the loss weighting ablation study that we promised in our original rebuttal. In our revised paper, we include an additional experiment in Appendix D Table 6 that shows the performance comparison of different weighting scalar applied to the likelihood distillation loss terms ($L_{div} + L_{D}$). In particular, we conduct the experiment with Shortcut-Distill-F2D2 on CIFAR10, with the modified loss function $L = L_{VM} + L{u} + \lambda(L_{div} + L_{D})$, and we also include the results below:
> > >
> > > | Scaling          | 8 Steps NLL | 8 Steps FID | 4 Steps NLL | 4 Steps FID | 2 Steps NLL | 2 Steps FID | 1 Step NLL | 1 Step FID |
> > > |------------------|------------:|------------:|------------:|------------:|------------:|------------:|-----------:|-----------:|
> > > | Shortcut-Distill |      -11.42 |        5.01 |      -26.82 |        5.41 |      -57.72 |        7.13 |    -119.42 |      12.75 |
> > > | 10               |        1.47 |       11.66 |        0.82 |       11.23 |       -0.07 |       10.11 |      -2.19 |      18.27 |
> > > | 1                |     **3.12**|     5.68|     **2.87**|     5.96|     **2.38**|     7.35|    **1.92**|   13.76|
> > > | 0.1              |      _2.59_ |      _5.01_ |      _2.00_ |      _5.33_ |      _1.78_ |      _7.01_ |     _0.68_ |     _13.04_|
> > > | 0.01             |        2.04 |     **4.87**|        1.51 |     **5.28**|        0.59 |     **6.76**|      -1.68 |   **12.70**|
> > >
> > > The table above shows the Shortcut-Distill-F2D2 results with $\lambda  = 0.01, 0.1, 1, 10$, with $\lambda = 1$ equivalent to our original setting and the performance of Shortcut-Distill without F2D2 as reference. As we can observe, adding an additional loss weighting can potentially help F2D2 achieve a better sweet spot in balancing high sample quality and accurate likelihood estimation: While our original equal weighting loss ($\lambda = 1$) obtains the most calibrated likelihood, it slightly penalizes the FID scores. Similarly, significantly down-weighting ($\lambda  = 0.01$) further improves FID but loses the ability to produce accurate likelihood. However, when choosing an appropriate $\lambda$ (in particular when $\lambda = 0.1$), this additional weighting can enable the model to produce both FID scores that match or even slightly improve the original Shortcut-Distill performance, and relatively calibrated log-likelihood estimation.
> > >
> > > This experiment opens the door to many intriguing future research directions that can potentially further improve the performance of our method. These include more fine-grained loss weighting scheme and developing training schedules similar to the ones proposed in shortcut models and MeanFlow, where the flow matching objective and the self-consistency objective are optimized in separate portions of the training.
> > >
> > > We sincerely thank the reviewer for this helpful suggestions and we hope this experiment resolves their concerns regarding this matter. Please feel free to comment if there is any other question we can answer. Thank you again for your time and efforts that went into reviewing our paper!

---

### Official Review · Reviewer_tED2 · 2025-10-31

**Soundness:** 2
**Presentation:** 3
**Contribution:** 2
**Rating:** 6
**Confidence:** 4

**Summary:**

This paper introduces a technique to add log probability estimates to popular shortcut distillation techniques. They achieve this by considering the combined sample and log-density differential equation and writing down additional loss terms for the latter part in terms of the divergence of the instantaneous velocity. They specify these terms for two popular frameworks: shortcut models and MeanFlow. The authors then proceed to evaluate their method on CIFAR10 and Imagenet64, reporting NLL and FID for their method and baselines based on both MeanFlow and shortcut models. The authors also present an experiment doing self-guidance for improved FID.

**Strengths:**

- The paper, especially up to the the experiments section is very well written, with easy to follow math that is presented with consistent notation throughout.
- I think the topic is interesting, with potentially important untapped potential of current generative models
- The method seems sound, and derivations (where applicable) seem correct

**Weaknesses:**

- My main concern with the paper is that the contribution seems somewhat limited, while it is to the best of my knowledge true that this method has not been applied before, it seems like a relatively straightforward extension of earlier shortcut methods.
- My other concern is that the experiments present some confusing results, and for a paper targeting log-likelihood, there is a lot of focus on FID in the provided tables, and not a lot of focus on log-likelihood. Other than the small experiment with self-guidance, there are also no experiments tailored specifically to investigate the quality of the approximation of the log-likelihood
- Continuing on the experimental evaluation: it seems strange to me to evaluate the dataset NLL to the dataset NLL of an expert, since it is comparing averages (i.e. it is a summary statistic). It would in my opinion make more sense to compare likelihoods directly on samples, and calculate a mean of errors, such as the mean squared error or mean absolute error. In the current format, individual samples could have arbitrarily large errors while the error of the mean could still be small.
- I would have hoped to see an experiment where the log-likelihood is more of a first-class member to see the effect of downstream performance. The authors mention for example the PPO family of methods in their introduction. It could also be instructive to add a toy-experiment in 1D or 2D where the ground-truth log density function is available.

**Questions:**

- The authors don't seem to to address the question whether the log-density still matches the log-density of the distribution. It seems to me that while the original distribution is is slow to sample from and the log-density similarly slow, at least there is theguarantee that the log-density is linked to the samples of the distribution. In this particular case, both parts (log-density and samples) are approximated, and it seems to me that there is no guarantee that these approximations match. Can the authors comment on whether the approximate log-density calculated is still a relevant number compared to the approximate distribution?
- What makes NLL predictions "invalid" (Tab 1. and Tab 2. gray fields)? Is this an arbitrary threshold that the authors chose or is there something else that makes them invalid?
- For CIFAR10, why do the authors underperform MeanFlow in some cases, and outperform it in other cases? Since the method operates on a separate head, their method should be fairly independent of performance, can the authors elaborate?
- For CIFAR10, it seems that the MeanFlow models produce better FID as the number of samples decreases. This is contrary to what I would expect, and also contrary to what is reported in the original paper. Can the authors elaborate why this would be? Can the authors answer the same question for MeanFlow-F2D2?

---

> ### Author Response · Authors · 2025-11-21
> **Author Rebuttal**
>
> We thank the reviewer for their thoughtful comments and their recognition of our work. We would like to answer their concerns and questions below.
>
> 1. **Contribution & novelty:** We thank the reviewer for noticing the simplicity of our approach. While flow map provides a unifying perspective and our method may seem natural in retrospect, we would like to emphasize that recognizing divergence can be jointly distilled alongside velocity is non-obvious, as evidenced by its absence in prior works despite significant active research on few-step sampling. We are the first diffusion/CNF-based distillation method that enables few-step likelihood evaluations and sampling simultaneously.
>
>     Likelihood evaluation is also a fundamental problem in generative modeling, and enables a range of applications, many of which we hope to explore in future work. Given the proliferation of distillation methods and prior lack of clarity about how these may be rigorously derived, demonstrating likelihood distillation within the flow map framework is a significant conceptual contribution that we see missing in the literature. Furthermore, to the best of our knowledge, we are the first to prove MeanFlow as an instantiation of the Eulerian flow map, which is another important contribution.
>
> 2. **Why did we include FID comparison:** Our evaluations include both sample quality and likelihood accuracy to demonstrate our ability to achieve good performance in both simultaneously. FID comparison also reveals a key failure mode of prior flow map methods: despite their ability to generate high quality samples, they fail to capture the true density and render invalid likelihood predictions. Our method on the other hand, is able to provide well calibrated likelihood estimations without sacrificing the sample quality.
>
> 3. **Experiments featuring likelihood evaluation:** Thank you for this suggestion. We agree that experiments featuring likelihood evaluation with analytically tractable ground truth can significantly strengthen our paper. Therefore, we have added Section 5.4 with 2D checkerboard experiments and compared our method against flow matching and vanilla shortcut model. As shown in Figure 3, flow matching and shortcut model completely fail at performing few-step likelihood evaluations while our method can achieve accurate results even with 1 NFE.
>
> 4. **Per-sample error for NLL:** Thank you for raising this important point. While we report BPD because it is the standard evaluation metric for NLL in the literature, we completely agree with the reviewer that per-sample error is a better metric to evaluate calibration quality. We have added Table 7 in our paper and below to show the mean absolute per-sample error in BPD w.r.t. the teacher's prediction. As we can observe, the F2D2 variants lower the error by 7-171x in comparison to the baselines. This evaluation directly verifies that our F2D2 can produce sample-level calibrated likelihood estimations that are suitable for practical applications, not just matching the summary statistics.
> | NFE                 | 8     | 4     | 2     | 1      |
> |-----------------------|-------|-------|-------|--------|
> | Shortcut              | 15.11 | 30.97 | 62.85 | 126.58 |
> | Shortcut-Distill      | 14.47 | 29.79 | 60.61 | 121.99 |
> | Meanflow              | 12.04 | 24.20 | 49.42 | 100.99 |
> | Shortcut-F2D2         | 0.52  | 0.55  | 1.01  | 3.08   |
> | Shortcut-Distill-F2D2 | **0.41**  | **0.50**  | **0.90**  | 1.69   |
> | MeanFlow-F2D2         | 1.62  | 1.78  | 1.79  | **0.59**   |
>
> 5. **What makes NLL predictions invalid:** We mark NLL predictions that are less than 0 BPD as invalid, which implies that the model assigns greater than 1 probability to the 8-bit images and violates probability axioms.
>
> 6. **Why do F2D2 affect FID:** While architecturally F2D2 only requires an additional prediction head, our method still requires end-to-end training for the backbone features to be sufficiently representative for both sampling and likelihood predictions. This joint optimization can influence FID in either direction, nevertheless we find that the changes in raw FID from F2D2 are rather modest, and it is not our claim that F2D2 as an auxiliary loss should be used for the sole purpose of improving FID. Rather, F2D2 naturally enables likelihood evaluation and, at times, FID improvement via test-time guidance.
>
> 7. **MeanFlow test-time scaling issue:** We have also noticed this counter-intuitive behavior. We use the pre-trained CIFAR10 checkpoint from the MeanFlow official PyTorch repository, which appears to be optimized for one-step generation only and therefore has degraded performance when scaled to multi-step. MeanFlow-F2D2 inherits the same behavior because we initialize its parameters from the pre-trained MeanFlow checkpoint. This observation actually motivated our self-guidance algorithm, where we demonstrate how tractable likelihood enables sample quality improvements with more inference-time computes.

---

### Meta-Review · Area_Chair_bazF · 2025-12-12

**Summary:**

This paper addresses an important  limitation of flow-matching models: the inability to obtain fast likelihood estimates and samples. The authors propose  a conceptually simple, well-motivated solution to this problem, which offers a  mechanism for jointly distilling both sampling trajectories and cumulative divergences for likelihood evaluations.
The empirical evaluation is generally solid, though somewhat limited. In particular, the comparison to ground-truth likelihoods  is restricted to a single toy example, which is not fully sufficient for assessing the practical reliability of the approximate log-likelihoods in real applications (e.g. in science). Nonetheless, the presented results demonstrate clear improvements over baselines in both sampling quality and the validity/calibration of few-step likelihood estimates.
Reviewer opinions largely positive.  There was one negative review that focused primarily on experimental scope and missing baselines. In my assessment, these concerns were addressed adequately in the rebuttal, and they do not undermine the core contribution.
Overall, the paper is well written, tackles a meaningful problem, and provides a compelling and broadly useful solution. On balance, I recommend acceptance.

**Reviewer Concerns:**

see meta review

**Reviewer Scores:**

I focused on the paper (see meta-review), not on trying to play theory-of-mind with the reviewers.

---

### Decision · Program_Chairs · 2026-01-26

Accept (Poster)